# Next Semantic Scale Prediction via Hierarchical Diffusion Language Models

Cai Zhou[1,*]    Chenyu Wang[1,*]    Dinghuai Zhang[2,3,*]    Shangyuan Tong[1]    Yifei Wang[1]
Stephen Bates[1,†]    Tommi Jaakkola[1,†]

[1]Massachusetts Institute of Technology  [2]Microsoft Research  [3]Mila - Quebec AI Institute
{caiz428, wangchy}@mit.edu  dinzhang@microsoft.com

## Abstract

In this paper we introduce Hierarchical Diffusion Language Models (HDLM) – a novel family of discrete diffusion models for language modeling. HDLM builds on a hierarchical vocabulary where low-level tokens with detailed semantics are surjectively mapped to high-level tokens with coarse-grained meanings. In the forward process, each token is independently perturbed to its higher-level ancestor with more abstract semantics according to the scheduler, while in the reverse process the model progressively predicts the next, more detailed semantics. Taken together, HDLM provides a general time-varying next semantic scale prediction process for language modeling. We derive closed-form expressions for the diffusion Evidence Lower Bound (ELBO), and show that HDLM can be implemented in a flexible manner while including the existing MDLM as a special case. We also propose practical training techniques based on the insights. Extensive text generation experiments validate the effectiveness of HDLM, which demonstrates consistently lower validation and generative perplexity than baselines.

## 1 Introduction

Autoregressive language models [1, 19, 36] are current state-of-the-art methods for language generation. However, the next-token prediction scheme fundamentally constrains their ability to revise previously generated tokens. Alternative paradigms such as diffusion models [31, 13, 32] have hence aroused researchers' interest due to their capability for progressive denoising and refinement. Discrete diffusion models, inherently compatible with the discrete nature of language, have recently gained popularity [20, 30, 26]. There are mainly two types of discrete diffusion model with different forward processes: uniform and masked (see [20] for more detailed descriptions). Nevertheless, both types have several fundamental limitations. For masked discrete diffusion, (i) all masked tokens have the same mask embeddings, lacking rich semantics; (ii) it cannot self-correct the tokens already generated, reintroducing the same non-revisability limitation of autoregressive models. For uniform discrete diffusion, it uniformly perturbs each token with a random token in the forward process. Some consequent drawbacks include (i) the same token serves as noise with high probability in the noisy stage but becomes semantically meaningful when it is decoded, resulting in inconsistency and confusion in the semantics; (ii) empirically, uniform discrete diffusion does not perform well, partially due to the higher difficulty in denoising from a more complex distribution.

Generalized Interpolating Discrete Diffusion (GIDD) [34] partially addresses these limitations by establishing a general framework for a broader class of processes, especially the interpolation between uniform and masked diffusion. GIDD combines both masking and uniform noise via a defined mixing rate. However, it is notable that the noisy tokens - including the masked tokens and uniformly

---

*Equal contribution. †Equal senior supervision.

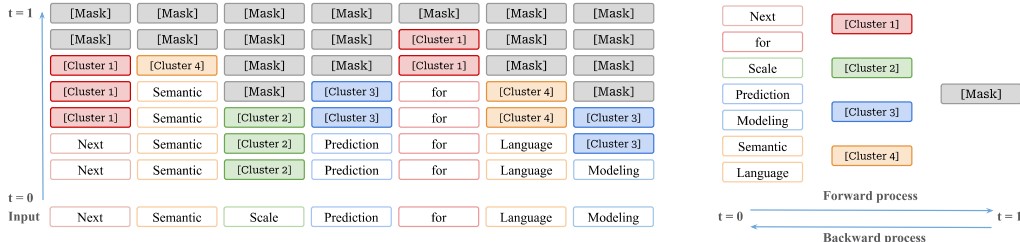

Figure 1: Hierarchical Diffusion Language Model (three hierarchies are shown). *(Left)* When training HDLM, word tokens transit to higher hierarchies independently with more abstract semantics. In the reverse process, the model learns to predict the lower level tokens. *(Right)* The vocabularies of three hierarchies (word, cluster, mask) and the illustration of next semantic scale prediction.

perturbed tokens - still lack rich semantics due to their discrete nature. In addition, the self-correction ability of GIDD comes only from the uniform noise, which actually harms the performance compared to masked diffusion.

To maximize the advantages of diffusion models, namely arbitrary order generation and progressive self-refinement, we introduce Hierarchical Diffusion Language Model (HDLM), a general and flexible framework for discrete diffusion language modeling via time-varying next-scale prediction. Analogously to the next-scale prediction in visual autoregressive models (VAR) [33], which generate image patch tokens from coarse to fine scales (lower to higher resolutions), we introduce semantic hierarchies into language tokens (Figure 1). In between clean word tokens and the mask token, we introduce intermediate hierarchies consisting of additional vocabularies. Each token in a lower hierarchy has more detailed semantics than those in a higher hierarchy, and there exists a surjective mapping from each lower hierarchy (fine grain with larger vocabulary size) to its adjacent higher hierarchy (coarse grain with smaller vocabulary size). In the forward diffusion process, each token is progressively turned into its mapping in the next higher hierarchy scale-by-scale at varying diffusion timesteps according to the designed scheduler. In the reverse process, the model learns to progressively predict more detailed tokens (i.e., the next scale of semantics) until the original words are recovered. Remarkably, in both forward and reverse processes, different tokens in a sentence may be in different hierarchies at a diffusion timestep, thus we denote our hierarchical discrete diffusion as **time-varying next-scale prediction** for language modeling, where the scale is in terms of the semantics of each token instead of spatial resolutions. Intuitively, this paradigm is consistent with the fact that some tokens are easier to predict than others, thus less denoising timesteps are needed to determine their detailed meanings in the next fine-grained hierarchies. The flexible and adaptive denoising process inherently takes advantage of the arbitrary order generation capability of diffusion models, improving the potential to surpass autoregressive language modeling.

Hierarchical discrete diffusion has several advantages over masked and uniform discrete diffusion. First, both mask and uniform noises lack rich semantics and will confuse the model, thus we introduce intermediate hierarchies. In comparison, the intermediate hierarchies can be viewed as partially masked tokens with high-level semantics, which are inherently more expressive than the single mask token or random tokens. Next, the uncertainty contained in the vague semantics of the coarse hierarchies allows for more flexibility in decoding, which improves the decoding accuracy and provides the possibility of self-refinement; i.e., errors in previous decisions can be mitigated to some extent, while masked discrete diffusion cannot correct decoded tokens at all. Finally, compared to the inconsistency between the semantics of clean tokens and the uniform random noises in uniform discrete diffusion, the semantics in hierarchical discrete diffusion are consistent and coherent, making it easier to accurately denoise in a progressive manner.

Time-varying next-scale prediction is a novel and principled scheme for language modeling. Despite the simplicity of its intuition, we establish rigorous theories for our hierarchical discrete diffusion based on the continuous-time Markov chain (CTMC) framework. Our hierarchical discrete diffusion is a special CTMC that restricts the transitions only to occur between adjacent hierarchies. We derive closed-form diffusion Evidence Lower Bound (ELBO) for hierarchical discrete diffusion training, which has clear interpretations. On the practical side, we show that the design space of hierarchical discrete diffusion is flexible, compatible with both GIDD and MDLM types of training and sampling both theoretically and heuristically. Code is available at https://github.com/zhouc20/HDLM.

## 2 Preliminary

Discrete diffusion models generalize the continuous diffusion framework to discrete state spaces $\mathcal{Z}$. Specifically, given an initial discrete data point $X \in \mathcal{Z}$ drawn from a data distribution $q_0(X)$, these models gradually degrade this data point through a forward Markov chain $Z_0, \ldots, Z_T$, where each state $Z_t$ evolves according to a transition kernel $q_{t|s}(Z_t|Z_s)$, starting from $Z_1 = X$ and eventually arriving at a prior distribution $p_T(Z_T)$, which is typically easy to sample from. We consider continuous-time Markov Chain in this paper, and select $T = 1$. The core task is then to train a model to approximate the reverse transitions of this Markov chain, effectively enabling it to reverse the degradation and recover original data points from samples drawn from the prior distribution.

D3PM [3] introduces a Markov forward process $q$ that acts independently on each token $q_{t|s}(z_t|z_s) = \text{Cat}(z_t; Q_{t|s}\mathbf{z}_s)$, where Cat denotes categorical distribution, and we use $\mathbf{z}_s$ to denote one-hot encoding vectors of $z_s$, namely the set of $\{\mathbf{x} \in \{0,1\}^{|V|}, \sum_{i=0}^{|V|} \mathbf{x}_i = 1\}$. With the noise schedule chosen in [30], the marginal of $z_t$ is $q(z_t|x) = \text{Cat}(z_t; \alpha_t\mathbf{x} + \beta_t\mathbf{m})$ where $\beta_t = 1 - \alpha_t$ and $\mathbf{m}$ denotes the one-hot vector of a given mask token.

Generalized Interpolating Discrete Diffusion (GIDD) [34] establishes a general framework for forward processes with flexible mixing schedules. In particular, the forward process is a Markov chain with marginal transitions that can be written as a linear interpolation of categorical distributions between a distribution $\boldsymbol{\pi}_t$ and data distribution $\mathbf{x}$:

$$q_t(z_t|x) = \text{Cat}(z_t; \alpha_t\mathbf{x} + \beta_t\boldsymbol{\pi}_t) \tag{1}$$

where $\mathbf{x}$ is the one-hot encoding of the data $x$, $\beta_t = 1 - \alpha_t$ where $0 \le \alpha_t \le 1$, and $\boldsymbol{\pi}_t$ can be any probability distribution that changes smoothly over time. Masked diffusion is a special case of GIDD where $\boldsymbol{\pi}_t = \mathbf{m}$. By defining the mixing rate and mixing distribution as follows, GIDD is the combination of masking and uniform noise:

$$\alpha_t = \frac{1-t}{C_t}, \beta_t\boldsymbol{\pi}_t = \frac{t}{C_t}\mathbf{m} + \frac{c_t}{C_t}\mathbf{1} \tag{2}$$

where $c_t = Bt^{\frac{\gamma}{2}}(1-t)^{\frac{\gamma}{2}}, C_t = 1 + Nc_t$ where $N$ is the vocabulary size, and $B$ is a constant chosen so that the desired uniform token ratio is reached. The marginal forward distribution is therefore

$$q_t(z_t|x) = \frac{1}{C_t}((1-t)\mathbf{x} + t\mathbf{m} + c_t\mathbf{1}) \tag{3}$$

However, it is notable that the noisy tokens - including the masked tokens and uniformly perturbed tokens - still lack rich semantics due to their discrete nature. In addition, the self-correction ability of GIDD comes only from the uniform noise, which actually harms the performance compared to masked diffusion.

## 3 Hierarchical Diffusion Language Model

We now introduce Hierarchical Diffusion Language Model, a family of discrete diffusion models with a next semantic scale prediction scheme. We build our derivations on hierarchical discrete diffusion models, providing flexible models that are compatible with prior work. See Appendix A for proofs.

### 3.1 Forward Process

We adopt the continuous-time Markov Chain (CTMC) as the mathematical framework for discrete diffusion models. Instead of transporting a token directly from clean data $\mathbf{x}$ to masks $\mathbf{m}$ (or vice versa) as MDLM does [30, 26], we introduce an intermediate hierarchy with expanded vocabulary to the Markov chain: $\mathbf{x} \to \mathbf{c} \to \mathbf{m}$, The higher level tokens have disjoint support and low-level tokens with detailed semantics are surjectively mapped to high-level tokens with coarse-grained meaning. Here $\mathbf{c} = \mathbf{c}(\mathbf{x}) = \Gamma\mathbf{x}$ is a predefined function of $\mathbf{x}$ (e.g., a clustering of $\mathbf{x}$), where $\Gamma \in \mathbb{R}^{|C| \times |V|}$ is the matrix with $\Gamma(i, j) = 1$ if and only if $c(x_j) = c_i$ and 0 otherwise. The vocabulary and the mapping functions of the hierarchies can be either hard-coded or learnable (e.g., via DeepSets [39]). In our implementation, we pre-define the mappings by clustering the word token embeddings of a pre-trained model given a number of clusters; details of our "semantic lustering algorithm" are available in Appendix C.

By introducing this hierarchical decoding procedure, the intermediate states can be viewed as partially decoded, partially masked tokens, which enables richer semantics of the masks with moderate uncertainty. The marginal distribution of the forward process can therefore be written as

$$q_t(z_t|x) = \text{Cat}(z_t; \alpha_t \mathbf{x} + \beta_{t,c}\mathbf{c}(\mathbf{x}) + \beta_{t,m}\mathbf{m}) \tag{4}$$

where $\beta_{t,c} + \beta_{t,m} = \beta_t := 1 - \alpha_t$. This can be viewed as a natural extension of GIDD where

$$\boldsymbol{\pi}_t(\mathbf{x}) = \text{Cat}(\pi_t; \frac{\beta_{t,c}}{\beta_{t,c} + \beta_{t,m}}\mathbf{c}(\mathbf{x}) + \frac{\beta_{t,m}}{\beta_{t,c} + \beta_{t,m}}\mathbf{m}) \tag{5}$$

A significant difference in our hierarchical forward process is that the mixing distribution $\boldsymbol{\pi}_t$ is a function of clean data $\mathbf{x}$, resulting fundamentally distinct CTMC properties (analyzed as follows) as well as the new training ELBO (Section 3.3).

**CTMC with block conditional transitions.** We use the notation $\mathbf{z}_t \in \mathbb{R}^{|V|+|C|+1}$ to denote the one hot representation of $z_t$, which is the concatenation of word-, cluster- and mask-level vectors. Denote $P_{t|s}$ as the cumulative transition probabilities from state $z_s$ to state $z_t$ at times $s < t$, namely $q_{t|s}(z_t|z_s) = \text{Cat}(z_t; P_{t|s}\mathbf{z_s})$, where the $z_s$-th column in $P_{t|s}$ represents the probability when starting from state $z_s$. Denote the time-inhomogeneous generator matrix (also called the forward transition rate matrix) as $Q_t$ satisfying $\frac{d\mathbf{z}_t}{dt} = Q_t^\top \mathbf{z}_t$, which contains instantaneous transition rates. The Kolmogorov forward and backward equations for CTMC state:

$$\frac{\partial P_{t|s}}{\partial t} = Q_t^\top P_{t|s}, \quad \frac{\partial P_{t|s}}{\partial s} = -P_{t|s}Q_s^\top, \quad P_{s|s} = I \tag{6}$$

**Proposition 1** (Time-inhomogeneous generator and cumulative conditional transition matrix of HDLM). *The time-inhomogeneous generator matrix of HDLM is*

$$Q_t = \begin{bmatrix} \frac{\alpha_t'}{\alpha_t}I_{|V|} & -\frac{\alpha_t'}{\alpha_t}\Gamma^\top & 0 \\ 0 & \frac{\alpha_t'+\beta_{t,c}'}{\beta_{t,c}}I_{|C|} & -\frac{\alpha_t'+\beta_{t,c}'}{\beta_{t,c}}\Xi^\top \\ 0 & 0 & 0 \end{bmatrix} \tag{7}$$

*where $\Xi = \mathbf{1}^{1\times|C|}$. The cumulative conditional transition matrix of HDLM is:*

$$P_{t|s} = \begin{pmatrix} \frac{\alpha_t}{\alpha_s}I_{|V|} & 0 & 0 \\ \left(\int_s^t \frac{-\alpha_u'}{\alpha_s}\frac{\beta_{t,c}}{\beta_{u,c}}\,du\right)\Gamma & \frac{\beta_{t,c}}{\beta_{s,c}}I_{|C|} & 0 \\ \left(1 - \frac{\alpha_t}{\alpha_s} + \int_s^t \frac{\alpha_u'}{\alpha_s}\frac{\beta_{t,c}}{\beta_{u,c}}\,du\right)\Gamma\Xi & \left(1 - \frac{\beta_{t,c}}{\beta_{s,c}}\right)\Xi & 1 \end{pmatrix} \tag{8}$$

Proposition 1 suggests that our hierarchical discrete diffusion process corresponds to a CTMC with a block forward transition rate matrix $Q_t$ and has the mask as an absorbing state.

**Generalized processes with stochastic perturbations and multiple hierarchies.** At inference time, diffusion models denoise based on predictions of previous steps. The accumulation of error may result in OOD contexts compared to those seen during training. To mitigate this potential gap between training and sampling, we introduce an additional cluster token corruption mechanism into the forward process. Denoting the probability that $x$ transits to the correct cluster $c(x)$ as $0 < \xi \le 1$, i.e., $\mathbb{P}_t(c = c(x)) = \xi\beta_{t,c}, \mathbb{P}_t(c \ne c(x)) = (1 - \xi)\beta_{t,c}$, we assume that a word token will be corrupted into an inaccurate cluster token $c' \ne c(x)$ with probability $\frac{1-\xi}{N_c-1}\beta_{t,c}$, where $N_c$ is the number of clusters. When $\xi = 1$, it reduces to the standard HDLM introduced above. We hence have the following noise distribution,

$$\beta_t\boldsymbol{\pi}_t = \xi\beta_{t,c}\mathbf{c}(x) + \sum_{\mathbf{c}\backslash\mathbf{c}(x)} \frac{1}{N_c-1}(1-\xi)\beta_{t,c}\mathbf{c} + \beta_{t,m}\mathbf{m} \tag{9}$$

This perturbation mechanism helps to train the model to predict correct word tokens even from incorrect cluster tokens or within inaccurate contexts with corrupted clusters, allowing the model to

robustly "self-correct" in a principled manner. This intuition is strongly supported by the experimental result that the generative perplexity of HDLM with $\xi < 1$ is 60% smaller than the baselines, demonstrating the strong generalization and self-correction capability of HDLM.

Our framework also generalizes to an arbitrary number of hierarchical levels, not limited to one intermediate hierarchy. Denote the word tokens as the 0-th hierarchy and the masks as the $n$-th, then in the forward process with $n - 1$ intermediate hierarchies, the tokens follow the Markov chain $\mathbf{x} \to \mathbf{c}_1 \to \cdots \to \mathbf{c}_{n-1} \to \mathbf{m}$. Correspondingly, the model progressively decodes tokens from higher-level hierarchies with more abstract semantics to lower-level ones in the inference time. Multiple hierarchy levels allow for sophisticated control over the generation process, yet may require a more careful design of the sampling algorithm. In this work, we provide theoretical foundations for HDLM with arbitrary hierarchies, while leaving experimental implementations of more intermediate levels as future work. More details of the stochastic perturbation mechanism and HDLM with arbitrary hierarchy levels, including their conditional transition matrices, closed-form ELBOs and further discussions, are available in Appendix A.2.

## 3.2 Reverse Process

For a noisy input $z_t$ at timestep $t$, the model prediction of distribution over the word-level vocabulary is $\mathbf{x}_\theta(z_t, t) \in \mathbb{R}^{|V|}$, which is shortened as $\mathbf{x}_\theta$ when not causing confusion. We adopt the commonly used reverse process in previous work [3, 34] as follows,

$$p_\theta(z_s|z_t) = q_{t|s}(z_t|z_s)\frac{q_s(z_s|\mathbf{x}_\theta)}{q_t(z_t|\mathbf{x}_\theta)} \tag{10}$$

Here $q_t(\cdot|\mathbf{x}_\theta) := \sum_{i:x_i \in V} \mathbf{x}_\theta[i]q_t(\cdot|x_i)$, where $\mathbf{x}_\theta[i]$ is the predicted probability of token $x_i$. Furthermore, we can prove that the backward rate matrix $\hat{Q}_t^\theta$ of our hierarchical forward process with block conditional transition matrix shares the same relation with the forward rate as GIDD [34].

$$\hat{Q}_t^\theta(z_t, z_s) = -\delta_{z_s, z_t} \sum_{z'} Q_t(z', z_t)\frac{q_t(z'|\mathbf{x}_\theta)}{q_t(z_t|\mathbf{x}_\theta)} + Q_t(z_s, z_t)\frac{q_t(z_s|\mathbf{x}_\theta)}{q_t(z_t|\mathbf{x}_\theta)} \tag{11}$$

For convenience, we omit $\theta$ and use $\hat{Q}_t$ to denote $\hat{Q}_t^\theta$ while not causing confusion.

## 3.3 Training

**Closed-form CT-ELBO of HDLM.** We now derive closed-form expressions for diffusion Evidence Lower Bound (ELBO) of HDLM based on the forward and backward rate matrix of its hierarchical process. For arbitrary noise distribution $\boldsymbol{\pi}_t$ and schedules $\alpha_t, \beta_t$, we have ELBO as follows [34].

**Lemma 2** (Proposition H.4 in [34]). *For any CTMC diffusion process with marginals $q_t(z_t|x)$, forward rate $Q_t(z_s, z_t)$, and backward rate $\hat{Q}_t(z_t, z_s)$, the continuous-time ELBO is*

$$\log p(x) \geq \mathbb{E}_{t, z_t}\Big[ \sum_{z_s \neq z_t} Q_t(z_s, z_t)\frac{q_t(z_s|x)}{q_t(z_t|x)} \log \frac{\hat{Q}_t(z_t, z_s)q_t(z_t|x)}{Q_t(z_s, z_t)q_t(z_s|x)} + \hat{Q}_t(z_t, z_t) - Q_t(z_t, z_t) \Big] + C \tag{12}$$

$$= \mathbb{E}_{t, z_t}\Big[ \sum_{z_s \neq z_t} Q_t(z_s, z_t)\frac{q_t(z_s|x)}{q_t(z_t|x)} \log \frac{q_t(z_s|\mathbf{x}_\theta)q_t(z_t|x)}{q_t(z_t|\mathbf{x}_\theta)q_t(z_s|x)} - \sum_{z'} Q_t(z', z_t)\frac{q_t(z'|\mathbf{x}_\theta)}{q_t(z_t|\mathbf{x}_\theta)} \Big] + C$$

*where $t \sim \mathcal{U}(0, 1)$ and $C = \mathbb{E}_{q_0(z_0|x)}[\log p(x|z_0)] - D_{KL}(q_1(z_1|x) \| p_1(x_1))$.*

One could easily verify that $C = 0$ for HDLM and MDLM, hence will be omitted afterwards. Now for our standard hierarchical process (with one intermediate hierarchy and without stochastic perturbations), plugging in the forward rate matrix and working through some algebra yields the following main result for HDLM:

**Theorem 3** (Closed-form CT-ELBO for HDLM with hierarchical CTMC diffusion process and block conditional transition)**.**

$$\log p(x) \geq \mathbb{E}_{t,z_t}\left[\delta_{z_t,c}\frac{-\alpha'_t}{\beta_{t,c}}\log\left(\frac{p_\theta(x)}{\sum\limits_{x':\Gamma x'=\Gamma x} p_\theta(x')}\right) + \delta_{z_t,m}\frac{\beta'_{t,m}}{\beta_{t,m}}\log\left(\sum\limits_{x':\Gamma x'=\Gamma x} p_\theta(x')\right)\right]$$

$$-\underbrace{\mathbb{E}_{t,z_t}\left[\frac{\alpha'_t}{\alpha_t}\delta_{z_t,x} + \frac{\beta'_{t,c}}{\beta_{t,c}}\delta_{z_t,c} + \frac{\beta'_{t,m}}{\beta_{t,m}}\delta_{z_t,m}\right]}_{=0} \quad (13)$$

$$= \mathbb{E}_{t,z_t}\left[\delta_{z_t,c}\frac{-\alpha'_t}{\beta_{t,c}}\mathbf{x}^\top\log\frac{\mathbf{x}_\theta\odot(\Gamma^\top\Gamma\mathbf{x})}{\mathbf{x}_\theta^\top\Gamma^\top\Gamma\mathbf{x}} + \delta_{z_t,m}\frac{\beta'_{t,m}}{\beta_{t,m}}(\Gamma\mathbf{x})^\top\log(\Gamma\mathbf{x}_\theta)\right] \quad (14)$$

$$= -\mathbb{E}_{t,z_t}\left[\delta_{z_t,c}w_{t,c}\mathrm{CE}(\mathbf{x},\frac{\mathbf{x}_\theta\odot(\Gamma^\top\Gamma\mathbf{x})}{\mathbf{x}_\theta^\top\Gamma^\top\Gamma\mathbf{x}}) + \delta_{z_t,m}w_{t,m}\mathrm{CE}(\Gamma\mathbf{x},\Gamma\mathbf{x}_\theta)\right] \quad (15)$$

*where $p_\theta(x_i) = \mathbf{x}_\theta[i]$ is the predicted probability of token $x_i$, $\odot$ is the Hadamard (element-wise) product, and*

$$w_{t,c} = \frac{-\alpha'_t}{\beta_{t,c}}, \; w_{t,m} = \frac{\beta'_{t,m}}{\beta_{t,m}} \quad (16)$$

Interestingly, Equation (13) can be interpreted as a combination of two cross-entropy (CE) losses corresponding to different hierarchies. The first term applies to each cluster token, requiring the model to classify within the set where word tokens are mapped to the cluster. This can be viewed as if restricting the predictions of word-level tokens within the known cluster and dropping the probabilities outside the cluster; thus $\mathbf{x}_\theta$ is element-wise multiplied by $\Gamma^\top\Gamma\mathbf{x}$ and normalized by $\mathbf{x}_\theta^\top\Gamma^\top\Gamma\mathbf{x}$, implying that the model only needs to predict word tokens within the cluster. The second term applies to each mask token, which is equivalent to a cluster-level classification in which predicting the word tokens within the correct cluster suffices.

**Analysis of training ELBO.** We continue with a more in-depth analysis of the HDLM ELBO. It can be proved that the expectations of the weights for both the token-level and cluster-level cross-entropy loss are invariant to the schedule.

**Proposition 4** (Invariance of both token-level and cluster-level loss weights)**.**

$$\mathbb{E}_{t,z_t}[\delta_{z_t,c}w_{t,c}] = \mathbb{E}_t[-\alpha'_t] = 1 \quad (17)$$
$$\mathbb{E}_{t,z_t}[\delta_{z_t,m}w_{t,m}] = \mathbb{E}_t[\beta'_{t,m}] = 1 \quad (18)$$

Proposition 4 is an intriguing neat property that is a generalization of the invariant results in MDLM [15, 26]. This implies that while different schedules may result in various weights $w_{t,c}$ and $w_{t,m}$ - hence enabling the flexibility in emphasizing different stages of the diffusion process, the expectation of each loss weight w.r.t. timesteps and samples is always fixed to 1, which is reasonable. Despite the same expectations of weights, HDLM can actually achieve lower cross-entropy loss (implying higher single-step denoising quality) compared with MDLM, thanks to the cluster tokens to be predicted or in the context providing more information than mask tokens.

Proposition 3 suggests a standard training loss for HDLM based on the strict ELBO:

$$\mathcal{L}(\mathbf{x},\mathbf{x}_\theta,t) = \mathbb{E}_{t,z_t\sim q_t(z_t|x)}\left[\delta_{z_t,c}w_{t,c}\mathrm{CE}(\mathbf{x},\frac{\mathbf{x}_\theta\odot(\Gamma^\top\Gamma\mathbf{x})}{\mathbf{x}_\theta^\top\Gamma^\top\Gamma\mathbf{x}}) + \delta_{z_t,m}w_{t,m}\mathrm{CE}(\Gamma\mathbf{x},\Gamma\mathbf{x}_\theta)\right] \quad (19)$$

Consequently, we further show that MDLM is a special case of our HDLM where there is only one cluster token. Despite being intuitive by construction, we can prove that HDLM ELBO reduces to MDLM's, which further validates that our hierarchical discrete diffusion is a generic framework compatible with previous work but provides more design space.

**Remark 1.** *MDLM is a special case of HDLM where there is only one cluster whose embedding is the same as the mask.*

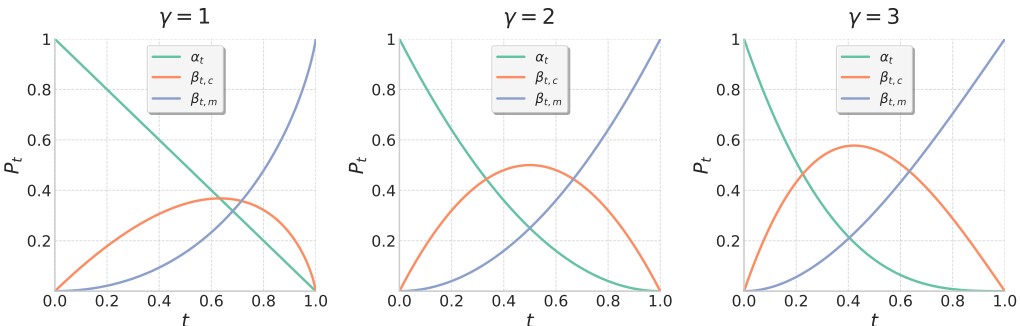

Figure 2: Marginal probabilities of the example forward processes with $\alpha_t = (1-t)^\gamma$ and $\gamma = 1, 2, 3$.

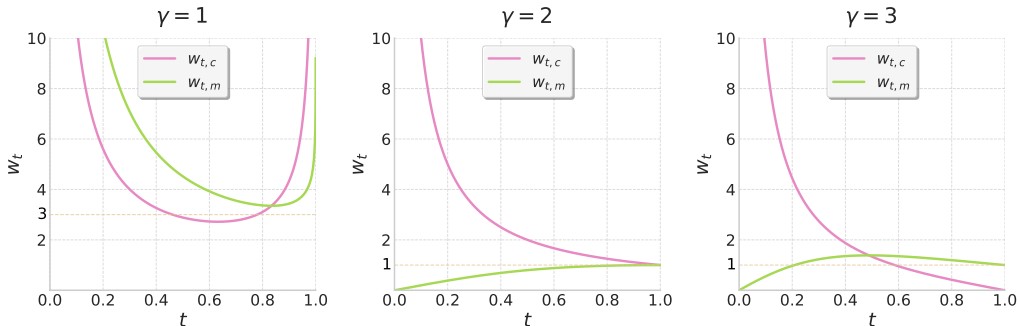

Figure 3: Cross entropy weights of the example forward processes with $\alpha_t = (1-t)^\gamma$ and $\gamma = 1, 2, 3$.

### 3.4 Example Processes

We now give some examples of forward processes. While the schedules are arbitrary, we consider the following transition process. When a token becomes a cluster token at time $\tau$, it will become a mask token at time $t_{m,\tau} \sim \mathcal{U}(\tau, 1)$. Thus the rate of cluster token and mask token follow,

$$\beta_{t,c} = \int_0^t \frac{1-t}{1-\tau}(-\mathrm{d}\alpha_\tau), \ \ \beta_{t,m} = \int_0^t \frac{t-\tau}{1-\tau}(-\mathrm{d}\alpha_\tau) \tag{20}$$

With $\alpha_t = (1-t)^\gamma$, we have the following general expression (visualized in Figure 2 for $\gamma = 1, 2, 3$):

$$\beta_{t,c} = \begin{cases} \frac{\gamma}{\gamma-1}(1-t-(1-t)^\gamma), \gamma \neq 1 \\ -(1-t)\ln(1-t), \gamma = 1 \end{cases} ; \ \ \beta_{t,m} = \begin{cases} \frac{1}{\gamma-1}(\gamma t + (1-t)^\gamma - 1), \gamma \neq 1 \\ t + (1-t)\ln(1-t), \gamma = 1 \end{cases} \tag{21}$$

Therefore, the weights $w_{t,c}, w_{t,m}$ can be computed as (visualized in Figure 3):

$$w_{t,c} = \begin{cases} (\gamma-1)\frac{(1-t)^{\gamma-2}}{1-(1-t)^{\gamma-1}}, \gamma \neq 1 \\ \frac{1}{-(1-t)\ln(1-t)}, \gamma = 1 \end{cases} ; \ \ w_{t,m} = \begin{cases} \frac{\gamma(1-(1-t)^{\gamma-1})}{\gamma t + (1-t)^\gamma}, \gamma \neq 1 \\ \frac{-\ln(1-t)}{t+(1-t)\ln(1-t)}, \gamma = 1 \end{cases} \tag{22}$$

### 3.5 Improved Techniques for HDLM Training and Sampling

Based on our theoretical insights, we continue to analyze the HDLM training and sampling processes, proposing more practical techniques to improve the model.

**Gradient of cluster-level loss.** Compared with MDLM, the cluster-level cross entropy loss in HDLM is additional. However, the loss at the cluster level would encourage predictions within the same clusters but not exactly the target word token $x$, causing the gradient to hold against optimizing

the cross entropy at the token level. In particular, consider the negative gradient of the cluster-level loss w.r.t. $\mathbf{x}_\theta(z_t, t)$,

$$-\nabla_{\mathbf{x}_\theta}\left(\delta_{z_t,m}w_{t,m}\text{CE}(\Gamma\mathbf{x}, \Gamma\mathbf{x}_\theta(z_t, t))\right) = \delta_{z_t,m}w_{t,m}\Gamma^\top(\Gamma\mathbf{x} \odot \frac{1}{\Gamma\mathbf{x}_\theta}) \tag{23}$$

This suggests that the gradient descent would encourage $\mathbf{x}_\theta$ predicting all the word tokens within the cluster $\{x' : c(x') = c(x)\}$ even if $x'$ are not the correct word token $x$. Intuitively, cluster prediction is an "easier" task, but actually benefits the model through an easy-to-hard curriculum and multi-stage decomposition: the model is allowed to make (moderate) mistakes in the earlier mask-to-cluster stage, as long as it can be corrected in the later cluster-to-token stage. To verify this hypothesis, we introduce a *hard training mode*, where we replace the cluster-level cross-entropy with the token-level cross-entropy (within the whole vocabulary) during training, leading to a harder task:

$$\mathcal{L}_{\text{Hard}}(\mathbf{x}, \mathbf{x}_\theta, t) = \delta_{z_t,c}w_{t,c}\text{CE}(\mathbf{x}, \frac{\mathbf{x}_\theta \odot (\Gamma^\top\Gamma\mathbf{x})}{\mathbf{x}_\theta^\top\Gamma^\top\Gamma\mathbf{x}}) + \delta_{z_t,m}w_{t,m}\text{CE}(\mathbf{x}, \mathbf{x}_\theta) \tag{24}$$

It is straightforward that $\text{CE}(\mathbf{x}, \mathbf{x}_\theta) \geq \text{CE}(\Gamma\mathbf{x}, \Gamma\mathbf{x}_\theta)$, so hard training mode actually utilizes a looser bound of the elbo as the training loss; the calculation of evaluation elbo remains the same. As shown in Table 5, the hard training mode indeed negatively affects the performance slightly, suggesting the advantage of progressively denoising and robustness of the original HDLM formulation.

**Auxiliary loss.** We apply the reverse process derived from the Bayesian perspective, so the model always outputs the word token prediction $\mathbf{x}_\theta(z_t, t)$ for all inputs at different noise levels, and the cluster-level prediction is derived through the Bayesian posterior $\Gamma\mathbf{x}_\theta$ instead of using another classifier head to directly predict the logits of clusters. As an ablation, we append an auxiliary cluster-level classifier head to the model which outputs $\mathbf{c}_\theta(z_t, t)$. Together with the main network, the auxiliary classifier is trained with the original elbo and an auxiliary cluster cross-entropy loss $\text{CE}(\Gamma\mathbf{x}, \mathbf{c}_\theta(z_t, t))$. We observe mild performance drop when using this direct cluster prediction instead of the Bayesian posterior (Table 5), demonstrating the advantage of self-consistent elbo training and posterior inference.

**Flexible loss weights.** We also find that clipping the large and extremely small loss weights based on heuristics help in stabilizing optimization and improving performance. Actually, such weight manipulation techniques can be commonly found across broad diffusion literature [14, 15, 34]. Intuitively, these weights can be thought of as hyperparameters over the noise schedule, which control how much effort the model devotes to the corresponding tasks during training. A careful balance, ideally through empirical tryouts, could often lead to better results [21].

**Force transition in decoding.** When predicting the word token from a cluster token, the naive model prediction $\mathbf{x}_\theta$ has logits for all tokens even outside the cluster. To keep consistent with the forward process, we can optionally restrict the model to decode the cluster tokens only to the word tokens within the clusters, which we call *force transition* (within the cluster). We adopt force transition decoding strategy by default and find it consistently helpful; see ablations for more discussion.

## 4 Experiments

### 4.1 Experimental Setup

In our experiments, we focus on language modeling, training the proposed HDLM algorithm on the widely used OpenWebText (OWT) [8] dataset following [34]. We adopt the settings in prior work [26, 34, 20, 30] in terms of architecture and scale - DiT [24] of small (170M) and base (425M) sizes with GPT-2 [25] tokenizer on 131B tokens of training data. We take 100000 data out of 8013769 data as the validation set. Following [34], we use a context length of 512 tokens and do not use sentence packing. We evaluate the trained models via both validation perplexity (Valid. PPL) and generative perplexity (Gen. PPL), which are standard metrics in recent literature. Other details and additional results are deferred to Appendix B.

Table 1: Validation perplexity and generative perplexity on OWT. The numbers of trained tokens are also reported for fair comparison. Our small models consistently surpass other discrete diffusion counterparts, and our base model is able to match the performance of autoregressive models.

| Model | Train. toks. | Valid. PPL ($\downarrow$) | Gen. PPL ($\downarrow$) |
|---|---|---|---|
| GPT2 [25][†] | unk. | 23.40 | - |
| Llama110M (retrain.)[†] | 262B | 16.11 | - |
| SEDD [20][†] | 262B | $\leq$24.10 | - |
| MDLM-small [26] (reimpl.) | 131B | $\leq$27.39 | 163.7 |
| GIDD+-small [34] (reimpl.) | 131B | $\leq$25.82 | 170.2 |
| HDLM-small-64 | 131B | $\leq$23.36 | **144.2** |
| HDLM-small-128 | 131B | $\leq$**23.25** | 148.0 |
| HDLM-base-128 | 131B | $\leq$19.22 | 139.9 |

## 4.2 Main Results

For Valid. PPL, we report the perplexity on the OWT validation set. For Gen. PPL, we generate 256 samples with length 512 using 512 denoising steps, and leverage gpt2-large [25] as the reference model. We include both autoregressive models and the main discrete diffusion models [20, 26, 34] as baselines, and the results are adopted from the corresponding work (marked with [†]) or reproduced in our settings (noted with reimpl.) to keep consistency. An HDLM with $n$ clusters is denoted as HDLM-$n$, and we also attach their sizes. Our main results are shown in Table 1. Remarkably, HDLMs with small size outperform other discrete diffusion variants in terms of both validation perplexity and generative perplexity. Moreover, our base HDLM achieves 19.77 perplexity, surpassing or matching the autoregressive models.

Table 2: Ablations of HDLM with different numbers of clusters $n$. We report the validation perplexity and generative perplexity on OWT, and all models are trained for 131B tokens. HDLM with cluster numbers around 64 and 128 achieve the best performance, while $n = 1$ recovers MDLM performance.

| Model | $n$ | Valid. PPL ($\downarrow$) | Gen. PPL ($\downarrow$) |
|---|---|---|---|
| | 1 | $\leq$25.72 | 163.9 |
| | 2 | $\leq$25.79 | 160.2 |
| | 4 | $\leq$25.05 | 162.0 |
| | 8 | $\leq$24.57 | 146.6 |
| HDLM-small | 16 | $\leq$24.17 | 155.2 |
| | 32 | $\leq$23.83 | 154.6 |
| | 64 | $\leq$23.36 | **144.2** |
| | 128 | $\leq$**23.25** | 148.0 |
| | 256 | $\leq$23.65 | 150.4 |

## 4.3 Ablation Studies

We conduct comprehensive ablations on HDLM w.r.t. the number of clusters $n$ (Table 2), the perturbation probability $\xi$ (Table 3), the forward process noise schedule controlled by $\gamma$ (Table 4), and different training modes (Table 5). Without specifying, we use HDLM-small and set $n = 64, \xi = 1, \gamma = 1$ and standard training mode by default. Full results are presented in Appendix B.

One could observe in Table 2 that as the number of clusters increases, the validation PPL first increases and then decreases, suggesting an optimal and moderate range for suitable numbers of clusters: approximately the square root of the vocabulary size which divide the generation into two stages with approximately equal complexities. We also verify empirically that HDLM with one cluster recovers the performance of MDLM, which is consistent with our theoretical analysis.

As illustrated in Table 3, stochastic perturbations introduce harder denoising target in training, which improves the model's ability to deal with inaccurate contexts and thus benefits the real inference

Table 3: Ablations of HDLM-64 with different perturbation probability $\xi$. We report the validation perplexity and generative perplexity on OWT, and all models are trained for 131B tokens. All models are forced transition within the cluster, which significantly reduces the Gen. PPL when $\xi < 1.0$.

| Model | $\xi$ | Valid. PPL ($\downarrow$) | Gen. PPL ($\downarrow$) |
|---|---|---|---|
| | 1.0 | $\leq 23.36$ | 144.2 |
| HDLM-small-64 | 0.9 | $\leq 23.54$ | 69.76 |
| | 0.8 | $\leq 25.93$ | 54.15 |

procedure: the generative PPL reduces over $51\%$ with $\xi = 0.9$ and over $62\%$ with $\xi = 0.8$. Intriguingly, force transition decoding strategy is effective even for $\xi < 1$. Intuitively, force transition would lead to better performance when the cluster prediction is accurate, yet otherwise will increase the accumulation prediction error. Our empirical results imply that the robustness to inaccurate contexts may be more important than correcting the cluster tokens in-place. For reference, the Gen. PPL is $184.0$ for $\xi = 0.9$ without force transition.

# 5 Related Work

Researchers have been looking for alternatives of autoregressive models for text despite their great success, among which diffusion models are representatives. Early work in this line often leverage continuous diffusion [18] or combine diffusion with AR models [12, 10]. Building on foundational work of discrete diffusion [3], more recent work such as SEDD [20], MDLM [26], MD4 [30] demonstrate the effectiveness of discrete diffusion in text modeling in small-scale pretraining, which is also our focus. In particular, SEDD [20] learns the concrete score (as the ratio of marginal distributions), while MDLM [26] and MD4 [30] further simplify the training objective for the masking forward process. Larger-scale work LLaDA [23] and Dream [38] demonstrate the scaling potential of diffusion language models, matching state-of-the-art AR models with the same scales in practical math reasoning [41], coding [11], or multimodal tasks [17, 37]. Most of the aforementioned recent discrete diffusion language models are masked diffusion due to the simplicity and empirical effectiveness, yet at the cost of losing self-correction ability. While there are also discrete diffusion with uniform noises [29, 27], they typically underperform the discrete counterparts. GIDD [34] unifies the two types by establishing a general framework that combines both noises. Our paper, by contrast, introduce a novel noising procedure that is both conceptually simple and practically effective, while preserving the self-correction ability. [35, 9] further investigate the integration of discrete diffusion with AR models. Block Diffusion [2] explores another new paradigm to autoregressively generate text blocks and diffuse text within each block. Our work acts on a different direction, as the length of sequence is fixed but the semantic of each token is generated sequentially.

# 6 Conclusion

We introduce Hierarchical Diffusion Language Model (HDLM), a generic and flexible family of discrete diffusion model with hierarchical semantic structures. Intuitively, HDLM follows a "next semantic scale prediction" scheme to progressively decode high-level abstract tokens to low-level detailed tokens. Leveraging the continuous-time Markov chain framework, we elaborate on the forward and backward process of hierarchical discrete diffusion, derive strict diffusion training ELBO and formally demonstrate their properties. Based on theoretical insights and empirical observations, we propose design principles and practical training techniques for HDLM. Our extensive experiments demonstrate the strong language modeling capability of HDLM.

**Limitations and Broader Impact**

Hierarchical discrete diffusion is a novel and general framework for discrete data generation. However, in our implementation we only consider one intermediate layer, while extending to multiple hierarchies is natural. We also leave exploring applications of hierarchical discrete diffusion to other areas besides language modeling (e.g., image generation, biomolecule design, and decision making) as future work.

**Acknowledgment**

C. Z. and S. B. were partly supported by the ARPA-H ADAPT program. C. Z., C. W., S. T. and T. J. were partly supported by the Machine Learning for Pharmaceutical Discovery and Synthesis (MLPDS) consortium, the NSF Expeditions grant (award 1918839) Understanding the World Through Code, and the DSO Singapore grant on next generation techniques for protein ligand binding.

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

## A Proofs

### A.1 Proof and Analysis of Standard HDLM ELBO

**Proof of HDLM ELBO.** We start with proving our main result: the ELBO of standard HDLM.

**Theorem 5** (Closed-form CT-ELBO for HDLM with hierarchical CTMC diffusion process and block conditional transition, Theorem 3 in the main text)**.**

$$\log p(x) \geq \mathbb{E}_{t,z_t} \Big[ \delta_{z_t,c} \frac{-\alpha'_t}{\beta_{t,c}} \log \Big( \frac{p_\theta(x)}{\sum\limits_{x':\Gamma\mathbf{x}'=\Gamma\mathbf{x}} p_\theta(x')} \Big) + \delta_{z_t,m} \frac{\beta'_{t,m}}{\beta_{t,m}} \log \Big( \sum_{x':\Gamma\mathbf{x}'=\Gamma\mathbf{x}} p_\theta(x') \Big) \Big]$$

$$\underbrace{- \mathbb{E}_{t,z_t} \Big[ \frac{\alpha'_t}{\alpha_t} \delta_{z_t,x} + \frac{\beta'_{t,c}}{\beta_{t,c}} \delta_{z_t,c} + \frac{\beta'_{t,m}}{\beta_{t,m}} \delta_{z_t,m} \Big]}_{=0} \tag{25}$$

$$= \mathbb{E}_{t,z_t} \Big[ \delta_{z_t,c} \frac{-\alpha'_t}{\beta_{t,c}} \mathbf{x}^\top \log \frac{\mathbf{x}_\theta \odot (\Gamma^\top \Gamma \mathbf{x})}{\mathbf{x}_\theta^\top \Gamma^\top \Gamma \mathbf{x}} + \delta_{z_t,m} \frac{\beta'_{t,m}}{\beta_{t,m}} (\Gamma \mathbf{x})^\top \log(\Gamma \mathbf{x}_\theta) \Big] \tag{26}$$

$$= -\mathbb{E}_{t,z_t} \Big[ \delta_{z_t,c} w_{t,c} \mathrm{CE}(\mathbf{x}, \frac{\mathbf{x}_\theta \odot (\Gamma^\top \Gamma \mathbf{x})}{\mathbf{x}_\theta^\top \Gamma^\top \Gamma \mathbf{x}}) + \delta_{z_t,m} w_{t,m} \mathrm{CE}(\Gamma \mathbf{x}, \Gamma \mathbf{x}_\theta) \Big] \tag{27}$$

*where $p_\theta(x_i) = \mathbf{x}_\theta[i]$ is the predicted probability of token $x_i$, $\odot$ is the Hadamard (element-wise) product, and*

$$w_{t,c} = \frac{-\alpha'_t}{\beta_{t,c}}, \ w_{t,m} = \frac{\beta'_{t,m}}{\beta_{t,m}} \tag{28}$$

*Proof.* By Lemma 2, we plugin $Q_t$ in Equation (7) into Equation (12). We calculate the following Equation (*) for different $z_t \in \{x, c(x), m\}$ respectively:

$$\sum_{z_s \neq z_t} Q_t(z_s, z_t) \frac{q_t(z_s|x)}{q_t(z_t|x)} \log \frac{q_t(z_s|\mathbf{x}_\theta) q_t(z_t|x)}{q_t(z_t|\mathbf{x}_\theta) q_t(z_s|x)} - \sum_{z'} Q_t(z', z_t) \frac{q_t(z'|\mathbf{x}_\theta)}{q_t(z_t|\mathbf{x}_\theta)} \tag{*}$$

(i) $z_t = m$, only when $z_s = c(x)$ that $Q_t(z_s, z_t) \neq 0$ and $q_t(z_s|x) \neq 0$, thus the first term being nonzero; when $z' \in c$, the second term is nonzero.

$$* = \frac{-(\alpha'_t + \beta'_{t,c})}{\beta_{t,c}} \frac{\beta_{t,c}}{\beta_{t,m}} \log \Big( \frac{\beta_{t,c} \cdot \sum\limits_{x':\Gamma\mathbf{x}'=\Gamma\mathbf{x}} p_\theta(x')}{\beta_{t,m} \cdot 1} \frac{\beta_{t,m}}{\beta_{t,c}} \Big) - \frac{-(\alpha'_t + \beta'_{t,c})}{\beta_{t,c}} \frac{\beta_{t,c} \cdot \sum\limits_{z' \in c} \sum\limits_{x':\Gamma\mathbf{x}'=\mathbf{z}'} p_\theta(x')}{\beta_{t,m} \cdot 1} \tag{29}$$

$$= \frac{\beta'_{t,m}}{\beta_{t,m}} \Big[ \log \Big( \sum_{x':\Gamma\mathbf{x}'=\Gamma\mathbf{x}} p_\theta(x') \Big) - 1 \Big] \tag{30}$$

where we utilize the facts that $-(\alpha'_t + \beta'_{t,c}) = \beta'_{t,m}$, and $\sum_{z' \in c} \sum_{x':\Gamma\mathbf{x}'=\mathbf{z}'} p_\theta(x') = 1$.

(ii) $z_t = c(x)$, the first term is nonzero only when $z_s = x$, and the second term is nonzero if $z' \in \{\Gamma z' = c(x), z' \in V\}$ or $z' = c(x)$, therefore

$$* = -\frac{\alpha'_t}{\alpha_t} \frac{\alpha_t}{\beta_{t,c}} \log \Big( \frac{\alpha_t p_\theta(x)}{\beta_{t,c} \sum\limits_{x':\Gamma\mathbf{x}'=\Gamma\mathbf{x}} p_\theta(x')} \frac{\beta_{t,c}}{\alpha_t} \Big) - \Big[ -\frac{\alpha'_t}{\alpha_t} \frac{\alpha_t \sum\limits_{x':\Gamma\mathbf{x}'=\Gamma\mathbf{x}} p_\theta(x')}{\beta_{t,c} \sum\limits_{x':\Gamma\mathbf{x}'=\Gamma\mathbf{x}} p_\theta(x')} + \frac{\alpha'_t + \beta'_{t,c}}{\beta_{t,c}} \cdot 1 \Big] \tag{31}$$

$$= \frac{-\alpha'_t}{\beta_{t,c}} \log \Big( \frac{p_\theta(x)}{\sum\limits_{x':\Gamma\mathbf{x}'=\Gamma\mathbf{x}} p_\theta(x')} \Big) - \frac{\beta'_{t,c}}{\beta_{t,c}} \tag{32}$$

(iii) $z_t = x$, the first term is always zero since $Q_t(z_s, z_t) = 0$ for all $z_s \neq z_t$, and the second term is nonzero only when $z' = z_t = x$, hence

$$* = 0 - \frac{\alpha_t'}{\alpha_t} \cdot 1 = -\frac{\alpha_t'}{\alpha_t} \tag{33}$$

Combining all three categories, we have

$$* = \delta_{z_t,m} \frac{\beta_{t,m}'}{\beta_{t,m}} \Big[ \log \Big( \sum_{x':\Gamma x'=\Gamma x} p_\theta(x') \Big) - 1 \Big] + \delta_{z_t,c} \Big[ \frac{-\alpha_t'}{\beta_{t,c}} \log \Big( \frac{p_\theta(x)}{\sum\limits_{x':\Gamma x'=\Gamma x} p_\theta(x')} \Big) - \frac{\beta_{t,c}'}{\beta_{t,c}} \Big] + \delta_{z_t,x}\Big(-\frac{\alpha_t'}{\alpha_t}\Big) \tag{34}$$

where we simplify $\delta_{z_t,c(x)}$ to $\delta_{z_t,c}$ since the only cluster token that $z_t$ could be is $c(x)$. Taking the expectation yields,

$$\mathbb{E}_{t,z_t}[*] = \mathbb{E}_{t,z_t} \Big[ \delta_{z_t,m} \frac{\beta_{t,m}'}{\beta_{t,m}} \log \Big( \sum_{x':\Gamma x'=\Gamma x} p_\theta(x') \Big) + \delta_{z_t,c(x)} \frac{-\alpha_t'}{\beta_{t,c}} \log \Big( \frac{p_\theta(x)}{\sum\limits_{x':\Gamma x'=\Gamma x} p_\theta(x')} \Big) \Big] \tag{35}$$

$$- \mathbb{E}_{t,z_t} \Big[ \delta_{z_t,x} \frac{\alpha_t'}{\alpha_t} + \delta_{z_t,c} \frac{\beta_{t,c}'}{\beta_{t,c}} + \delta_{z_t,m} \frac{\beta_{t,m}'}{\beta_{t,m}} \Big] \tag{36}$$

The second term has an expectation of zero:

$$\mathbb{E}_t \Big[ \mathbb{E}_{z_t \sim q_t(z_t|x)} \Big( \delta_{z_t,x} \frac{\alpha_t'}{\alpha_t} + \delta_{z_t,c} \frac{\beta_{t,c}'}{\beta_{t,c}} + \delta_{z_t,m} \frac{\beta_{t,m}'}{\beta_{t,m}} \Big) \Big] \tag{37}$$

$$= \mathbb{E}_t \Big[ \alpha_t \frac{\alpha_t'}{\alpha_t} + \beta_{t,c} \frac{\beta_{t,c}'}{\beta_{t,c}} + \beta_{t,m} \frac{\beta_{t,m}'}{\beta_{t,m}} \Big] \tag{38}$$

$$= \int_0^1 (\alpha_t' + \beta_{t,c}' + \beta_{t,m}') \mathrm{d}t \tag{39}$$

$$= \alpha_t \Big|_0^1 + \beta_{t,c} \Big|_0^1 + \beta_{t,m} \Big|_0^1 \tag{40}$$

$$= -1 + 0 + 1 \tag{41}$$

$$= 0 \tag{42}$$

The first term can be simplified into matrix form which is equivalent to a weighted sum of two cross-entropy losses:

$$\mathbb{E}_{t,z_t} \Big[ \delta_{z_t,m} \frac{\beta_{t,m}'}{\beta_{t,m}} \log \Big( \sum_{x':\Gamma x'=\Gamma x} p_\theta(x') \Big) + \delta_{z_t,c} \frac{-\alpha_t'}{\beta_{t,c}} \log \Big( \frac{p_\theta(x)}{\sum\limits_{x':\Gamma x'=\Gamma x} p_\theta(x')} \Big) \Big] \tag{43}$$

$$= \mathbb{E}_{t,z_t} \Big[ \delta_{z_t,c} \frac{-\alpha_t'}{\beta_{t,c}} \mathbf{x}^\top \log \frac{\mathbf{x}_\theta \odot (\Gamma^\top \Gamma \mathbf{x})}{\mathbf{x}_\theta^\top \Gamma^\top \Gamma \mathbf{x}} + \delta_{z_t,m} \frac{\beta_{t,m}'}{\beta_{t,m}} (\Gamma \mathbf{x})^\top \log(\Gamma \mathbf{x}_\theta) \Big] \tag{44}$$

$$= -\mathbb{E}_{t,z_t} \Big[ \delta_{z_t,c} w_{t,c} \mathrm{CE}\Big(\mathbf{x}, \frac{\mathbf{x}_\theta \odot (\Gamma^\top \Gamma \mathbf{x})}{\mathbf{x}_\theta^\top \Gamma^\top \Gamma \mathbf{x}}\Big) + \delta_{z_t,m} w_{t,m} \mathrm{CE}(\Gamma \mathbf{x}, \Gamma \mathbf{x}_\theta) \Big] \tag{45}$$

where $p_\theta(x_i) = \mathbf{x}_\theta[i]$ is the predicted probability of token $x_i$, $\odot$ is the Hadamard (element-wise) product, and

$$w_{t,c} = \frac{-\alpha_t'}{\beta_{t,c}}, \ w_{t,m} = \frac{\beta_{t,m}'}{\beta_{t,m}} \tag{46}$$

We therefore conclude the CT-ELBO for HDLM:

$$\log p(x) \geq -\mathbb{E}_{t,z_t} \Big[ \delta_{z_t,c} w_{t,c} \mathrm{CE}\Big(\mathbf{x}, \frac{\mathbf{x}_\theta \odot (\Gamma^\top \Gamma \mathbf{x})}{\mathbf{x}_\theta^\top \Gamma^\top \Gamma \mathbf{x}}\Big) + \delta_{z_t,m} w_{t,m} \mathrm{CE}(\Gamma \mathbf{x}, \Gamma \mathbf{x}_\theta) \Big] \tag{47}$$

$\square$

The conclusion that MDLM is a special case of HDLM directly follows.

**Corollary 6.** *MDLM is a special case where there is only one cluster (which equals to the mask).*

*Proof.* We need to show that when the neural network parameters are identical, our MDLM ELBO reduces to MDLM. According to the HDLM ELBO in Theorem 3,

$$-\log p(x) \leq -\mathbb{E}_{t,z_t}\left[\delta_{z_t,c}w_{t,c}\text{CE}(\mathbf{x}, \frac{\mathbf{x}_\theta \odot (\Gamma^\top \Gamma \mathbf{x})}{\mathbf{x}_\theta^\top \Gamma^\top \Gamma \mathbf{x}}) + \delta_{z_t,m}w_{t,m}\text{CE}(\Gamma \mathbf{x}, \Gamma \mathbf{x}_\theta)\right] \quad (48)$$

Here all word tokens belong to the same cluster, i.e., $\Gamma = \mathbf{1}^{1\times|V|}$, thus it always holds that the second term is zero:

$$\text{CE}(\Gamma \mathbf{x}, \Gamma \mathbf{x}_\theta(z_t, t)) = 0 \quad (49)$$

For the first term in Equation (13), we have the following derivation:

$$\mathbb{E}_{t,z_t}\left[\delta_{z_t,c}w_{t,c}\text{CE}(\mathbf{x}, \frac{\mathbf{x}_\theta \odot (\Gamma^\top \Gamma \mathbf{x})}{\mathbf{x}_\theta^\top \Gamma^\top \Gamma \mathbf{x}})\right] = \mathbb{E}_{t,z_t}\left[\delta_{z_t,c}w_{t,c}\text{CE}(\mathbf{x}, \mathbf{x}_\theta)\right] = \mathbb{E}_t\left[-\alpha_t'\text{CE}(\mathbf{x}, \mathbf{x}_\theta)\right] \quad (50)$$

which is the same as the MDLM [26]. □

Actually, an interesting fact is that MDLM is also equivalent to HDLM with $|V|$ (vocabulary size) clusters. Hence, any cluster number in between $1$ and $|V|$ implies a generalized model with additional benefits compared with MDLM, and it is reasonable to observe an optimal cluster size (around $64$ to $128$ in our experiments).

**Analysis of the training ELBO for the example forward processes.** We now give more analysis on the example forward processes stated in the main text - note that these are only the schedules used in our experiments, while HDLM supports more flexible schedules. Consider the following transition process: when a token becomes a cluster token at time $\tau$, it will become a mask token at time $t_{m,\tau} \sim \mathcal{U}(\tau, 1)$. Then in the forward process, the rate of cluster token and mask token will be

$$\beta_{t,c} = \int_0^t \frac{1-t}{1-\tau}(-\text{d}\alpha_\tau) \quad (51)$$

$$\beta_{t,m} = \int_0^t \frac{t-\tau}{1-\tau}(-\text{d}\alpha_\tau) \quad (52)$$

With $\alpha_t = (1-t)^\gamma$, we have the following general expression:

$$\beta_{t,c} = \begin{cases} \frac{\gamma}{\gamma-1}(1-t-(1-t)^\gamma), \gamma \neq 1 \\ -(1-t)\ln(1-t), \gamma = 1 \end{cases} \quad (53)$$

$$\beta_{t,m} = \begin{cases} \frac{1}{\gamma-1}(\gamma t + (1-t)^\gamma - 1), \gamma \neq 1 \\ t + (1-t)\ln(1-t), \gamma = 1 \end{cases} \quad (54)$$

Therefore, the weights $w_{t,c}, w_{t,m}$ can be computed as

$$w_{t,c} = \frac{-\alpha_t'}{\beta_{t,c}} = \begin{cases} (\gamma-1)\frac{(1-t)^{\gamma-2}}{1-(1-t)^{\gamma-1}}, \gamma \neq 1 \\ \frac{1}{-(1-t)\ln(1-t)}, \gamma = 1 \end{cases} \quad (55)$$

$$w_{t,m} = \frac{\beta_{t,m}'}{\beta_{t,m}} = \begin{cases} \frac{\gamma(1-(1-t)^{\gamma-1})}{\gamma t + (1-t)^\gamma}, \gamma \neq 1 \\ \frac{-\ln(1-t)}{t + (1-t)\ln(1-t)}, \gamma = 1 \end{cases} \quad (56)$$

When the schedules follow the common practice that $\alpha_t$ is always decreasing w.r.t. $t$ and $\beta_{t,m}$ is increasing, we have $\alpha_t' \leq 0, \beta_{t,m}' \geq 0$, leading to non-negative weights:

$$w_{t,c} \geq 0, \ w_{t,m} \geq 0, \ \forall t \in [0,1] \quad (57)$$

While Proposition 4 guarantees that the expectations for token- and cluster-level loss weights are finite, their limits at $t \to 0_+$ and $t \to 1_-$ may not be finite. In particular,

$$\lim_{t \to 0_+} w_{t,c} \to +\infty, \text{ if } \lim_{t \to 0_+} \alpha'_t \neq 0 \tag{58}$$

$$\lim_{t \to 1_-} w_{t,c} \to +\infty, \text{ if } \lim_{t \to 1_-} \alpha'_t \neq 0 \tag{59}$$

$$\lim_{t \to 0_+} w_{t,m} \to +\infty, \text{ if } \lim_{t \to 0_+} \beta'_{t,m} \neq 0 \tag{60}$$

In typical schedules, these conditions tend to be satisfied - for instance, in the linear schedule $\alpha_t = 1 - t$ and $\alpha_t = -1$, resulting in infinite weights. However, it is reasonable for $w_{0,c}$ and $w_{0,m}$ to be infinite, since all the cluster and mask tokens must be fully decoded to clean word tokens when $t = 0$. Moreover, the dominators of the weights $\beta_{t,c}, \beta_{t,m}$ are exactly the occurrence probabilities of cluster/mask tokens in the forward process, thus we have

$$\mathbb{E}_{z_t \sim q_t(z_t|x)}[\delta_{z_t,c} w_{t,c}] = -\alpha'_t \tag{61}$$

$$\mathbb{E}_{z_t \sim q_t(z_t|x)}[\delta_{z_t,m} w_{t,m}] = \beta'_{t,m} \tag{62}$$

which are finite as long as the time derivatives of $\alpha_t$ and $\beta_{t,m}$ are finite as in common practice. The above analysis suggests a high degree of freedom in designing forward processes, as we obtain proper training ELBOs in most cases under mild conditions.

## A.2 Extended Forward Processes

**HDLM with stochastic perturbations.** We now derive the ELBO for the case with random perturbations to the cluster tokens (i.e., $\xi < 1$).

**Theorem 7** (Closed-form CT-ELBO for HDLM with stochastic perturbations).

$$\begin{aligned}
\log p(x) \geq \mathbb{E}_{t,z_t} \Bigg[ & \delta_{z_t,m} \frac{\beta'_{t,m}}{\beta_{t,m}} \Big( \xi \cdot \log \Big( p_\theta(c(x)) + \frac{1-\xi}{\xi(N_c - 1)}[1 - p_\theta(c(x))] \Big) \\
& + \frac{1-\xi}{N_c - 1} \cdot \sum_{z_s = c \setminus c(x)} \log \Big( \frac{\xi(N_c - 1)}{1 - \xi} p_\theta(z_s) + [1 - p_\theta(z_s)] \Big) \Big) \\
& + \delta_{z_t, c(x)} \frac{-\alpha'_t}{\beta_{t,c}} \log \frac{p_\theta(x)}{p_\theta(z_t) + \frac{1-\xi}{\xi(N_c - 1)}[1 - p_\theta(z_t)]} \\
& + \delta_{z_t, c \setminus c(x)} \frac{-\alpha'_t}{\beta_{t,c}} \log \frac{p_\theta(x)}{\frac{\xi(N_c - 1)}{1 - \xi} p_\theta(z_t) + [1 - p_\theta(z_t)]} \Bigg]
\end{aligned} \tag{63}$$

Here we use the notation $p_\theta(z_s) = \sum_{x_i:c(x_i)=z_s} p_\theta(x_i), z_s \in C$ to simplify the expression. To interpret the ELBO, denote $\eta = \frac{(N_c - 1)\xi}{1 - \xi}$, which is usually greater than 1 for reasonable $\xi$ values (since $\xi$ should not be much smaller than 1). Iin particular, when $\xi = 1, \eta \to \infty$. Then Equation (63) simplifies to:

$$\begin{aligned}
\log p(x) \geq \mathbb{E}_{t,z_t} \Bigg[ & \delta_{z_t,m} \frac{\beta'_{t,m}}{\beta_{t,m}} \xi \Big( \log \Big( (1 - \frac{1}{\eta}) p_\theta(c(x)) + \frac{1}{\eta} \Big) + \frac{1}{\eta} \sum_{z_s = c \setminus c(x)} \log \Big( (\eta - 1) p_\theta(z_s) + 1 \Big) \Big) \\
& + \delta_{z_t, c(x)} \frac{-\alpha'_t}{\beta_{t,c}} \log \frac{p_\theta(x)}{(1 - \frac{1}{\eta}) p_\theta(z_t) + \frac{1}{\eta}} + \delta_{z_t, c \setminus c(x)} \frac{-\alpha'_t}{\beta_{t,c}} \log \frac{p_\theta(x)}{(\eta - 1) p_\theta(z_t) + 1} \Bigg]
\end{aligned} \tag{64}$$

*Proof.* The inhomogeneous generator $Q_{t,\xi}$ can be obtained by replacing $\Gamma$ in Equation (7) by $\tilde{\Gamma}_\xi$, which is defined by substituting every 1 in $\Gamma$ with $\xi$ and every 0 with $\frac{1-\xi}{N_c - 1}$. We continue to calculate Equation (*).

(i) $z_t = x$, the calculation is the same as those in Theorem 3,

$$* = -\frac{\alpha'_t}{\alpha_t} \tag{65}$$

(ii) $z_t = c(x)$, which means the word token transits to a correct cluster token. Denote $p_\theta(c(x)) = \sum_{x':\Gamma x'=c(x)} p_\theta(x')$, then the second term in Equation (*) needs to sum over all $z' \in V$ and $z' = z_t = c(x)$:

$$-\sum_{z'} Q_t(z', z_t) \frac{q_t(z'|\mathbf{x}_\theta)}{q_t(z_t|\mathbf{x}_\theta)} = -\frac{\alpha'_t}{\alpha_t} \frac{\alpha_t \xi p_\theta(c(x)) + \alpha_t \frac{1-\xi}{N_c-1}(1-p_\theta(c(x)))}{\beta_{t,c} \xi p_\theta(c(x)) + \beta_{t,c} \frac{1-\xi}{N_c-1}(1-p_\theta(c(x)))} - \frac{\alpha'_t + \beta'_{t,c}}{\beta_{t,c}} \cdot 1 = -\frac{\beta'_{t,c}}{\beta_{t,c}}$$
(66)

The first term is zero when $z_s$ is any of cluster and mask tokens (since $Q_{t,\xi}(z_s, c(x)) = 0$ when $z_s \neq c(x)$, and zero for any word token that is not $x$ since $q_t(z_s|x) = 0$ when $z_s \in \{V\backslash x\}$. Thus the first term is

$$\sum_{z_s \neq z_t} Q_t(z_s, z_t) \frac{q_t(z_s|x)}{q_t(z_t|x)} \log \frac{q_t(z_s|\mathbf{x}_\theta)q_t(z_t|x)}{q_t(z_t|\mathbf{x}_\theta)q_t(z_s|x)}$$
(67)

$$= \frac{-\alpha'_t \xi}{\alpha_t} \frac{\alpha_t}{\beta_{t,c} \xi} \log \left( \frac{p_\theta(x)\alpha_t}{\xi p_\theta(c(x)) + \frac{1-\xi}{N_c-1}(1-p_\theta(c(x)))} \frac{\xi \beta_{t,c}}{\alpha_t} \right)$$
(68)

$$= \frac{-\alpha'_t}{\beta_{t,c}} \log \left( \frac{p_\theta(x)}{p_\theta(z_t) + \frac{1}{\eta}(1-p_\theta(z_t))} \right)$$
(69)

(iii) $z_t \in \{C\backslash c(x)\}$, which could only happen when $\xi < 1$. The second term in Equation (*) can be computed as:

$$-\sum_{z'} Q_t(z', z_t) \frac{q_t(z'|\mathbf{x}_\theta)}{q_t(z_t|\mathbf{x}_\theta)} = -\frac{\alpha'_t}{\alpha_t} \frac{\alpha_t \xi p_\theta(z_t) + \alpha_t \frac{1-\xi}{N_c-1}(1-p_\theta(z_t))}{\beta_{t,c} \xi p_\theta(z_t) + \beta_{t,c} \frac{1-\xi}{N_c-1}(1-p_\theta(z_t))} - \frac{\alpha'_t + \beta'_{t,c}}{\beta_{t,c}} \cdot 1 = -\frac{\beta'_{t,c}}{\beta_{t,c}}$$
(70)

Analogously, the first term is only nonzero when $z_s = x$,

$$\sum_{z_s \neq z_t} Q_t(z_s, z_t) \frac{q_t(z_s|x)}{q_t(z_t|x)} \log \frac{q_t(z_s|\mathbf{x}_\theta)q_t(z_t|x)}{q_t(z_t|\mathbf{x}_\theta)q_t(z_s|x)}$$
(71)

$$= \frac{-\alpha'_t}{\alpha_t} \frac{(1-\xi)}{N_c-1} \frac{\alpha_t}{\beta_{t,c} \frac{(1-\xi)}{N_c-1}} \log \left( \frac{p_\theta(x)\alpha_t}{\xi p_\theta(z_t) + \frac{1-\xi}{N_c-1}(1-p_\theta(z_t))} \frac{\frac{(1-\xi)}{N_c-1}\beta_{t,c}}{\alpha_t} \right)$$
(72)

$$= \frac{-\alpha'_t}{\beta_{t,c}} \log \left( \frac{p_\theta(x)}{\eta p_\theta(z_t) + (1-p_\theta(z_t))} \right)$$
(73)

(iv) $z_t = m$, the second term sums over all cluster tokens,

$$-\sum_{z'} Q_t(z', z_t) \frac{q_t(z'|\mathbf{x}_\theta)}{q_t(z_t|\mathbf{x}_\theta)} = \frac{\alpha'_t + \beta'_{t,c}}{\beta_{t,c}} \frac{\beta_{t,c} \sum_{z' \in C} \xi p_\theta(z') + \frac{1-\xi}{N_c-1}(1-p_\theta(z'))}{\beta_{t,m}} = -\frac{\beta'_{t,m}}{\beta_{t,m}}$$
(74)

where we leverage $\sum_{z' \in C} p_\theta(z') = 1$. For the first term, we need to sum over all cluster tokens whether $z_s = c(x)$ or not,

$$\sum_{z_s \neq z_t} Q_t(z_s, z_t) \frac{q_t(z_s|x)}{q_t(z_t|x)} \log \frac{q_t(z_s|\mathbf{x}_\theta)q_t(z_t|x)}{q_t(z_t|\mathbf{x}_\theta)q_t(z_s|x)}$$
(75)

$$= \frac{-(\alpha'_t + \beta'_{t,c})}{\beta_{t,c}} \left[ \frac{\beta_{t,c}\xi}{\beta_{t,m}} \log \left( \frac{(p_\theta(c(x))\xi + [1-p_\theta(c(x))]\frac{1-\xi}{N_c-1})\beta_{t,c}}{\beta_{t,m} \cdot 1} \frac{\beta_{t,m}}{\beta_{t,c}\xi} \right) \right.$$

$$\left. + \frac{\beta_{t,c}\frac{1-\xi}{N_c-1}}{\beta_{t,m}} \sum_{z_s=c\backslash c(x)} \log \left( \frac{(p_\theta(z_s)\xi + (1-p_\theta(z_s))\frac{1-\xi}{N_c-1})\beta_{t,c}}{\beta_{t,m} \cdot 1} \frac{\beta_{t,m}}{\beta_{t,c}\frac{1-\xi}{N_c-1}} \right) \right]$$
(76)

$$= \frac{\beta'_{t,m}}{\beta_{t,m}} \xi \left[ \log \left( p_\theta(c(x)) + \frac{1}{\eta}[1-p_\theta(c(x))] \right) + \frac{1}{\eta} \sum_{z_s=c\backslash c(x)} \log \left( \eta p_\theta(z_s) + (1-p_\theta(z_s)) \right) \right]$$
(77)

Combining all types, we have

$$\mathbb{E}_{t,z_t}[*] = \mathbb{E}_{t,z_t}\left[\delta_{z_t,m}\frac{\beta'_{t,m}}{\beta_{t,m}}\xi\left(\log\left((1-\frac{1}{\eta})p_\theta(c(x))+\frac{1}{\eta}\right)+\frac{1}{\eta}\sum_{z_s=c\backslash c(x)}\log\left((\eta-1)p_\theta(z_s)+1\right)\right)\right.$$

$$+\delta_{z_t,c(x)}\frac{-\alpha'_t}{\beta_{t,c}}\log\frac{p_\theta(x)}{(1-\frac{1}{\eta})p_\theta(z_t)+\frac{1}{\eta}}+\delta_{z_t,c\backslash c(x)}\frac{-\alpha'_t}{\beta_{t,c}}\log\frac{p_\theta(x)}{(\eta-1)p_\theta(z_t)+1}\Bigg]$$

$$+\mathbb{E}_{t,z_t}\left[\delta_{z_t,m}\frac{\beta'_{t,m}}{\beta_{t,m}}+\delta_{z_t,c(x)}\frac{\beta'_{t,c}}{\beta_{t,c}}+\delta_{z_t,c\backslash c(x)}\frac{\beta'_{t,c}}{\beta_{t,c}}+\delta_{z_t,x}\frac{\alpha'_t}{\alpha_t}\right] \tag{78}$$

For similar reason, one could easily obtain that the expectation of the bias term is zero, hence we conclude the proof. □

One can verify that when $\xi = 1$, the ELBO reduces to the original version without perturbations in Theorem 3. The ELBO actually encourages correct mask-to-cluster prediction (through the first term) and correct cluster-to-token prediction (through the second term), as well as out-of-cluster exploration when the current cluster state $z_t$ is inaccurate (through the dominator of the last term).

**HDLM with arbitrary hierarchy levels.** For consistency and simplicity, suppose the clean tokens are the $0$-th hierarchy, and the mask token is the $n$-th hierarchy. Denote $c_i(x)$ as the mapping of a clean token $x$ to the $i$-th hierarchy. Let $\alpha_{t,i}$ be the marginal schedule for the $i$-th hierarchy. We assume $\xi = 1.0$ for simplicity, then the ELBO is given as follows.

**Theorem 8** (Closed-form CT-ELBO for HDLM with arbitrary number of hierarchies).

$$\log p(x) \geq \mathbb{E}_{t,z_t}\left[\sum_{i=1}^n \delta_{z_t,i}\frac{-\sum_{j<i}\alpha'_{t,j}}{\alpha_{t,i}}\log\left(\frac{\sum_{x':c_{i-1}(x')=c_{i-1}(x)}p_\theta(x')}{\sum_{x'':c_i(x'')=c_i(x)}p_\theta(x'')}\right)\right] \tag{79}$$

*where $\delta_{z_t,i}$ is an indicator function that denotes whether $z_t$ is in the $i$-th hierarchy.*

*Proof.* We start with calculating the time-inhomogeneous generator for HDLM with $n$ hierarchies. By recursion method, we have

$$Q_{t,n} = \begin{bmatrix} \frac{\alpha'_{t,0}}{\alpha_{t,0}}I_{|V|} & -\frac{\alpha'_{t,0}}{\alpha_{t,0}}\Gamma_0^\top & 0 & \cdots & 0 & 0 \\ 0 & \frac{\alpha'_{t,0}+\alpha'_{t,1}}{\alpha_{t,1}}I_{|C_1|} & -\frac{\alpha'_{t,0}+\alpha'_{t,1}}{\alpha_{t,1}}\Gamma_1^\top & \cdots & 0 & 0 \\ \cdots & \cdots & \cdots & \cdots & \cdots & \cdots \\ \cdots & \cdots & \cdots & \cdots & \cdots & \cdots \\ 0 & 0 & 0 & \cdots & \frac{\sum_{j=0}^{n-1}\alpha'_{t,j}}{\alpha_{t,n-1}}I_{|C_{n-1}|} & -\frac{\sum_{j=0}^{n-1}\alpha'_{t,j}}{\alpha_{t,n-1}}\Gamma_{n-1} \\ 0 & 0 & 0 & \cdots & 0 & 0 \end{bmatrix} \tag{80}$$

where $\Gamma_i \in \mathbb{R}^{|C_{i+1}|\times|C_i|}$ maps tokens in the $i$-th hierarchy to the $i+1$-th hierarchy.

Analogously to the proof for Theorem 3, according to Lemma 2, plugin the above $Q_{t,n}$ into Equation (12) yields the following calculation:

(i) $i = 0$ (namely $z_t = x$), the first term in $(*)$ is zero, and the second term is nonzero only when $z' = z_t = x$, thus

$$* = -\frac{\alpha'_{t,0}}{\alpha_{t,0}} \tag{81}$$

(ii) $0 < i < n$ (namely $z_t = c_i(x)$ the mapping in the $i$-th intermediate hierarchy), only when $z_s = c_{i-1}(x)$ that the first term is nonzero, and only when $z' = z_t = c_i(x)$ or $z' \in \{z' \in C_{i-1}:$

$\Gamma_{i-1}\mathbf{z}' = \mathbf{z}_t\}$ the second term is nonzero, hence

$$* = \frac{-\sum_{j<i}\alpha'_{t,j}}{\alpha_{t,i-1}}\frac{\alpha_{t,i-1}}{\alpha_{t,i}}\log\Big(\frac{\alpha_{t,i-1}}{\alpha_{t,i}}\frac{\sum_{x':c_{i-1}(x')=c_{i-1}(x)}p_\theta(x')}{\sum_{x'':c_i(x'')=c_i(x)}p_\theta(x'')}\frac{\alpha_{t,i}}{\alpha_{t,i-1}}\Big)$$

$$- \Big(\frac{\sum_{j\le i}\alpha'_{t,j}}{\alpha_{t,i}}\cdot 1 + \frac{-\sum_{j<i}\alpha'_{t,j}}{\alpha_{t,i-1}}\cdot\frac{\alpha_{t,i-1}}{\alpha_{t,i}}\Big) \tag{82}$$

$$= \frac{-\sum_{j<i}\alpha'_{t,j}}{\alpha_{t,i}}\log\Big(\frac{\sum_{x':c_{i-1}(x')=c_{i-1}(x)}p_\theta(x')}{\sum_{x'':c_i(x'')=c_i(x)}p_\theta(x'')}\Big) - \frac{\alpha'_{t,i}}{\alpha_{t,i}} \tag{83}$$

(iii) $i = n$ (namely $z_t = m$), only when $z_s = c_{n-1}(x)$ that the first term is nonzero, and only when $z' \in \{z' \in C_{n-1} : \Gamma_{n-1}\mathbf{z}' = \mathbf{z}_t\}$ the second term is nonzero, therefore

$$* = \frac{-\sum_{j<n}\alpha'_{t,j}}{\alpha_{t,n-1}}\frac{\alpha_{t,n-1}}{\alpha_{t,n}}\log\Big(\frac{\alpha_{t,n-1}}{\alpha_{t,n}}\frac{\sum_{x':c_{n-1}(x')=c_{n-1}(x)}p_\theta(x')}{\sum_{x'':c_n(x'')=c_n(x)}p_\theta(x'')}\frac{\alpha_{t,n}}{\alpha_{t,n-1}}\Big)$$

$$- \Big(\frac{-\sum_{j<n}\alpha'_{t,n}}{\alpha_{t,n-1}}\cdot\frac{\alpha_{t,n-1}}{\alpha_{t,n}}\Big) \tag{84}$$

$$= \frac{-\sum_{j<n}\alpha'_{t,n}}{\alpha_{t,n}}\log\Big(\frac{\sum_{x':c_{n-1}(x')=c_{n-1}(x)}p_\theta(x')}{\sum_{x'':c_n(x'')=c_n(x)}p_\theta(x'')}\Big) - \frac{\alpha'_{t,n}}{\alpha_{t,n}} \tag{85}$$

where we again leverage $-\sum_{j<n}\alpha'_{t,j} = \alpha'_{t,n}$.

Combining all the terms, we have the constant bias term

$$\mathbb{E}_{t,z_t}\sum_{i=0}^{n}\delta_{z_t,c_i(x)}\frac{-\alpha'_{t,i}}{\alpha_{t,i}} = -\sum_{i=0}^{n}\int_0^1\alpha'_{t,i}\mathrm{d}t = 0 \tag{86}$$

We therefore have the remaining weighted sum of cross-entropy losses:

$$\log p(x) \ge \mathbb{E}_{t,z_t}\Big[\sum_{i=1}^{n}\delta_{z_t,i}\frac{-\sum_{j<i}\alpha'_{t,j}}{\alpha_{t,i}}\log\Big(\frac{\sum_{x':c_{i-1}(x')=c_{i-1}(x)}p_\theta(x')}{\sum_{x'':c_i(x'')=c_i(x)}p_\theta(x'')}\Big)\Big] \tag{87}$$

$\square$

The result shows that the ELBO of arbitrary hierarchies has a clear interpretation: it can be decomposed into a weighted combination of cross-entropy losses on each hierarchy, where the model learns to decode a token on the $i$-th hierarchy into the correct detailed token in the $i-1$-th hierarchy. We believe this is a novel and highly nontrivial theoretical contribution which could inspire more future work.

## B  Experimental Details

### B.1  Experimental Settings

We develop models based on the DiT architecture [24] and use the GPT2 [25] tokenizer. We train two model sizes, consistent with previous research [34, 26]: SMALL (containing 12 layers, 12 attention heads, and a hidden dimensions of 768, totaling 92.1M non-embedding parameters), and BASE (containing 24 layers, 16 attention heads, and a hidden dimensions of 1024, totaling 321.2M non-embedding parameters).

Following the setting of [34], all models are trained with a context size of 512 tokens and a batch size of 512 for 500k steps, resulting in a total of 131B training tokens. We utilize 8 NVIDIA A100/H100 80GB GPUs and employ mixed precision training in bf16 format. For optimization, we use the Adam optimizer [16] ($\beta = (0.9, 0.99)$, $\epsilon = 10^{-9}$) with a learning rate of $5 \times 10^{-4}$. The learning rate is warmed up linearly for the first 10k steps and then decayed using a cosine schedule to 10% of the initial learning rate. We apply a weight decay of 0.02 and gradient clipping to a norm of 1.0. We clip the largest value of loss weights $w_{t,m}, w_{t,c}$ to 2.0 or 10.0 in training for stable optimization, yet do not clip while evaluating the elbo for fair comparison.

To ensure training stability, all denominators in the loss and ELBO weights were clipped to $1 \times 10^{-4}$. For sequences exceeding 512 tokens, we randomly selected a 512-token window. Shorter sequences were padded to 512 tokens; these padding tokens were included in the loss calculation but excluded from the ELBO.

For downstream performance evaluation, we follow the practice in GIDD [34] and use the `lm-eval-harness`[2]. We incorporate a custom model that estimates per-token log-likelihoods using the ELBO. Our evaluation focuses on likelihood-based multiple-choice tasks, where the per-token likelihood is calculated across both the context and the completion, but not the padding.

However, it is remarkable that validation perplexity is not an effectiveness metric, as models with larger vocabulary sizes like ours naturally distribute probability mass across more tokens, leading to higher perplexity. Moreover, the elbo is only an upper bound of the exact likelihood, and our complex forward process potentially leads to loose bounds and thus not directly comparable. However, HDLM still achieve much lower perplexity than baselines with smaller vocabulary sizes, which further verify its superiority. Generative perplexity is also imperfect as it measures the alignment with the reference model, and a simple, repetitive sequence tend to have a small value. Despite these limitations, we still report Valid. PPL and Gen. PPL as common practice in recent literature.

## B.2 Additional Results

**Ablation studies.** In the main text, we already provide the ablation results of cluster number $n$ and stochastic perturbation $\xi$. The results w.r.t. the forward process schedule $\gamma$ and training mode are further presented here.

Table 4: Ablations of HDLM-$64$ with different schedules parameterized by $\gamma$. We report the validation perplexity and generative perplexity on OWT, and all models are trained for 131B tokens. Intuitively, larger $\gamma$ introduces harder forward processes with worse validation PPL correspondingly, but leads to better generation quality in full inference.

| Model | $\gamma$ | Valid. PPL ($\downarrow$) | Gen. PPL ($\downarrow$) |
|---|---|---|---|
| | 1.0 | $\leq$23.36 | 144.2 |
| HDLM-small-64 | 2.0 | $\leq$25.01 | 138.3 |
| | 3.0 | $\leq$130.32 | 135.9 |

Reported in Table 4 are the results for HDLMs with different forward process schedules controlled by $\gamma$. Intuitively, a larger $\gamma$ lead to a harder single-step denoising task for a given $t$, leading to larger validtion PPL. However, HDLM with larger $\gamma$ would decode less tokens in the initial stage and decode more tokens when the context becomes richer, which improves the generative perplexity (i.e., the real-inference performance is enhanced). This observation is consistent with other literature that decoding more tokens in the later stage (namely at small $t$) is helpful.

Table 5: Ablations of HDLM-$64$ with different training strategies. We report the validation perplexity and generative perplexity on OWT, and all models are trained for 131B tokens.

| Model | Train Mode | Valid. PPL ($\downarrow$) | Gen. PPL ($\downarrow$) |
|---|---|---|---|
| | - | $\leq$23.36 | 144.2 |
| HDLM-small-64 | Auxiliary | $\leq$24.36 | 151.9 |
| | Hard | $\leq$25.38 | 151.9 |
| | Auxiliary+Hard | $\leq$25.21 | 143.2 |

Results in Table 5 suggest that the harder training task would slightly hurt the performance. This validates that the next semantic scale prediction paradigm of HDLM effectively decomposes the generation process into stages with balanced difficulty, bringing performance gain.

---

[2]https://github.com/EleutherAI/lm-evaluation-harness.

Table 6: Performance on LM1B dataset.

| Model | Valid. PPL ($\downarrow$) |
|---|---|
| AR Transformer (33B tokens) | 22.32 |
| BERT-Mouth | $\leq 142.89$ |
| D3PM (absorb) | $\leq 76.90$ |
| Diffusion-LM | $\leq 118.62$ |
| DiffusionBert | $\leq 63.78$ |
| SEDD (33B tokens) | $\leq 32.79$ |
| MDLM (33B tokens) | $\leq 27.04$ |
| HDLM-64 (33B tokens) | $\leq \mathbf{26.95}$ |

**Results on LM1B.** We additionally pretrain our model on LM1B [5], a smaller dataset that is popular for text generation. Following [26], we set the sequence length to 128 but without sentence packing (as in [34]). We train HDLM for 1M steps, resulting in 33B tokens. The constant learning rate is set to $3 \times 10^{-4}$ with 2500 warmup steps, and other hyperparameters follow the settings in OWT pretraining. As shown in Table 6, HDLM is able to achieve better performance than SEDD and MDLM baselines. Notably, the improvements on OWT is more significant, suggesting the advantages of HDLM could be more obvious in difficult tasks.

Table 7: Zero-shot benchmark accuracy on downstream datasets.

| Size | Model | Train. toks. | ARC-e | ARC-c | BoolQ | PIQA | OBQA | WinoG. | Avg. |
|---|---|---|---|---|---|---|---|---|---|
| SMALL | GPT2 | unk. | **43.81** | 19.03 | 48.72 | **62.89** | 16.40 | **51.62** | 40.41 |
| | Llama (retrain,) | 262B | 40.53 | 25.51 | 46.21 | 62.73 | 28.40 | 50.57 | **42.35** |
| | MDLM | 262B | 30.98 | 23.63 | 50.52 | 54.13 | 28.00 | 49.41 | 39.44 |
| | GIDD (our eval impl.) | 131B | 26.73 | 24.83 | **51.10** | 51.85 | 27.20 | 49.49 | 38.53 |
| | HDLM-4 (ours) | 131B | 27.15 | **26.62** | 50.43 | 52.50 | **30.00** | 50.99 | 39.62 |
| | HDLM-32 (ours) | 131B | 26.73 | 25.17 | 49.97 | 51.03 | 24.00 | **51.78** | 38.11 |
| | HDLM-32-$\xi = 0.9$ (ours) | 131B | 28.07 | 25.94 | 50.61 | 53.21 | 27.00 | 51.07 | 39.32 |
| BASE | GIDD (our eval impl.) | 131B | 28.32 | 23.12 | 49.57 | 52.83 | 23.80 | 48.30 | 37.66 |
| | HDLM-256 (ours) | 131B | 28.91 | 25.09 | 50.18 | 55.33 | 26.20 | 51.85 | 39.59 |

**Downstream evaluation.** We evaluate the general performance of our language model for language understanding based on the commonly adopted lm-eval evaluation suite. We follow the benchmark suite choice of [34], which cover a diverse range of tasks regarding general language understanding and question answering capability and can convincingly showcase the performance comparison among different models. These include ARC [7] (both elementary and challenge subsets), BoolQ [6], HellaSwag [40], PIQA [4], OpenBookQA [22], and WinoGrande [28].

We demonstrate the results of this evaluation suite in Table 7. For small and base model sizes, we report the performance of HDLM-4 and HDLM-256 respectively. Our small model achieves the best on two datasets and highly competitive results on others, even in comparison with autoregressive models. The average accuracy also outperforms other diffusion models even trained for twice as long, verifying the strong generalization power of HDLM. Our large model is better than the GIDD counterpart in every dataset. Interestingly, increasing model size does not necessarily improve the performance as observed by [34], which we partially attribute to overfitting the pretraining dataset. We also observe that with the perturbed forward process, the model with $\xi < 1$ learns to distinguish inaccurate clusters in the denoising steps, which improves its self-correction ability. Compared to its counterpart whose $\xi = 1$, HDLM-32 with $\xi = 0.9$ is better in five out of six tasks, and is also superior in four tasks to HDLM-4, verifying the potential advantage of self-correction.

# C Details of the Semantic Clustering Algorithm

## C.1 Clustering algorithm of word embeddings

As mentioned in the main text, the clusters of the word tokens can be learned jointly with the discrete diffusion model or predefined. In our implementation, we predefine the mapping given the number of clusters with an enhanced K-means clustering algorithm, which we refer as Semantic Clustering. The algorithm determine the clusters according to the token embeddings of a pretrained model, for instance, a pretrained GIDD model [34] which is more compatible with our discrete diffusion. Semantic Clustering consists of a K-means++ initialization, a dynamic cluster size control mechanism (to keep the sizes of the clusters relatively balanced up to a use-defined tolerance), as well as efficient cluster assignment and centroids updates. We adopt the converged assignment to determine our word-to-cluster mapping (i.e., $\Gamma$), and in addition take the centroids of clusters as the initializations of cluster token embeddings in HDLM. Detailed algorithms are described in Algorithm 1 and Algorithm 2.

## C.2 Examples of Word Clusters

We experiment with $\#\text{clusters} = \{2, 4, 8, 16, 32, 64, 128, 256\}$, with pretrained GIDD model of both base and small sizes. The clustering algorithm generally converges in 1 minute. The overall semantic coherence lies between $0.3$ to $0.4$ for most settings, suggesting a reasonable correlation within the clusters (given the large vocabulary size and the high-dimensional nature of the word embeddings). The cluster sizes and example words of the clustering for $\#\text{clusters} = 32$ and GIDD-small is shown in Table 8 and Table 9 for reference, from which we can validate the effectiveness of the algorithm.

## C.3 Discussions

As one could imagine, the quality of the clustering method has considerable effect on the model performance. Hence, more sophisticated algorithm to predefine the hierarchical mapping is worth exploring, which is left as future work. The mapping function can even be extended to stochastic or implicit. Learning the hierarchies jointly with discrete diffusion training is also a feasible solution, where we could represent sets with deep models such as DeepSets [39]. Hierarchical clustering also provides an additional perspective for introducing multiple hierarchies into HDLM. We can show that our theoretical framework still applies to this case, and HDLM can potentially further benefit from more carefully designed or well-trained hierarchies without introducing computation overheads in practical training.

**Algorithm 1:** Semantic Clustering with Size Constraints (Part 1: Setup and initialization)

---

**Input:** $embeddings$, $cluster\_size$, $max\_iters$, $tolerance$, $min\_size\_ratio$,
$max\_size\_ratio$, $use\_cosine$
**Output:** $cluster\_ids$, $centroids$
$vocab\_size, embed\_dim \leftarrow$ Dimensions of $embeddings$;
**if** $use\_cosine = True$ **then**
   |   $normalized\_embeddings \leftarrow embeddings/\|embeddings\|$ ;    // Normalize vectors
**else**
   |   $normalized\_embeddings \leftarrow embeddings$;
**end**
$avg\_cluster\_size \leftarrow vocab\_size/cluster\_size$;
$min\_cluster\_size \leftarrow avg\_cluster\_size \times min\_size\_ratio$;
$max\_cluster\_size \leftarrow avg\_cluster\_size \times max\_size\_ratio$;
// Initialize centroids using K-means++
$centroids \leftarrow$
 InitializeKMeansPlusPlus($normalized\_embeddings, cluster\_size, use\_cosine$);
**Function** *InitializeKMeansPlusPlus(embeddings, cluster_size, use_cosine)*:
   |   $centroids \leftarrow$ zeros($cluster\_size, embed\_dim$);
   |   $first\_idx \leftarrow$ RandomInteger($0, vocab\_size - 1$);
   |   $centroids[0] \leftarrow embeddings[first\_idx]$;
   |   **if** $use\_cosine = True$ **then**
   |    |   $centroids[0] \leftarrow centroids[0]/\|centroids[0]\|$;
   |   **end**
   |   **for** $k = 1$ **to** $cluster\_size - 1$ **do**
   |    |   **if** $use\_cosine = True$ **then**
   |    |    |   $similarities \leftarrow embeddings \cdot centroids[: k]^T$;
   |    |    |   $max\_similarities \leftarrow$ max($similarities$, axis $= 1$);
   |    |    |   $distances \leftarrow 1 - max\_similarities$;
   |    |   **else**
   |    |    |   $distances \leftarrow$ ComputeMinDistances($embeddings, centroids[: k]$);
   |    |   **end**
   |    |   $probabilities \leftarrow distances^2 / \sum distances^2$;
   |    |   $next\_idx \leftarrow$ SampleFromDistribution($probabilities$);
   |    |   $centroids[k] \leftarrow embeddings[next\_idx]$;
   |    |   **if** $use\_cosine = True$ **then**
   |    |    |   $centroids[k] \leftarrow centroids[k]/\|centroids[k]\|$;
   |    |   **end**
   |   **end**
   |   **return** $centroids$;
**for** $iteration = 1$ **to** $max\_iters$ **do**
   |   // Assignment phase
   |   **if** $use\_cosine = True$ **then**
   |    |   $affinities \leftarrow$
   |    |    ComputeCosineSimilarities($normalized\_embeddings, centroids$);
   |    |   $cluster\_ids \leftarrow$ argmax($affinities$, axis $= 1$) ;      // Initial assignment
   |   **else**
   |    |   $distances \leftarrow$
   |    |    ComputeEuclideanDistances($normalized\_embeddings, centroids$);
   |    |   $cluster\_ids \leftarrow$ argmin($distances$, axis $= 1$) ;      // Initial assignment
   |   **end**
   |   $cluster\_sizes \leftarrow$ CountPointsInClusters($cluster\_ids, cluster\_size$);
   |   // Continue in Part 2...
**end**

---

**Algorithm 2:** Semantic Clustering with Size Constraints (Part 2: Update and Size Constraints)

```
// Continued from Part 1...
// Handle oversized clusters
```
$oversized\_clusters \leftarrow \{k \mid cluster\_sizes[k] > max\_cluster\_size\}$;
**foreach** $cluster\_idx \in oversized\_clusters$ **do**
  $excess\_count \leftarrow cluster\_sizes[cluster\_idx] - max\_cluster\_size$;
  $points\_in\_cluster \leftarrow \{i \mid cluster\_ids[i] = cluster\_idx\}$;
  **foreach** $point \in points\_in\_cluster$ **do**
    Calculate affinity/distance to current cluster;
    Find second best cluster and its affinity/distance;
    Calculate $reassignment\_cost$ for moving this point;
  **end**
  $points\_to\_move \leftarrow$ Select $excess\_count$ points with lowest $reassignment\_cost$;
  **foreach** $point \in points\_to\_move$ **do**
    Reassign $point$ to its second-best cluster;
  **end**
**end**
```
// Handle undersized clusters
```
$cluster\_sizes \leftarrow$ RecountClusters$(cluster\_ids, cluster\_size)$;
$undersized\_clusters \leftarrow \{k \mid cluster\_sizes[k] < min\_cluster\_size\}$;
**foreach** $cluster\_idx \in undersized\_clusters$ **do**
  $needed\_count \leftarrow min\_cluster\_size - cluster\_sizes[cluster\_idx]$;
  $candidate\_points \leftarrow \{i \mid cluster\_ids[i] \neq$
    $cluster\_idx$ and has high affinity to $cluster\_idx\}$;
  $points\_to\_move \leftarrow$ Select $needed\_count$ best candidates from $candidate\_points$;
  **foreach** $point \in points\_to\_move$ **do**
    $cluster\_ids[point] \leftarrow cluster\_idx$;
  **end**
**end**
```
// Update centroids
```
$new\_centroids \leftarrow$ zeros$(cluster\_size, embed\_dim)$;
**for** $k = 0$ **to** $cluster\_size - 1$ **do**
  $cluster\_mask \leftarrow \{i \mid cluster\_ids[i] = k\}$;
  **if** $|cluster\_mask| > 0$ **then**
    **if** $use\_cosine = True$ **then**
      $mean\_vector \leftarrow$ mean$(normalized\_embeddings[cluster\_mask], \texttt{axis} = 0)$;
      $new\_centroids[k] \leftarrow mean\_vector/\|mean\_vector\|$;
    **else**
      $new\_centroids[k] \leftarrow$ mean$(normalized\_embeddings[cluster\_mask], \texttt{axis} = 0)$;
    **end**
  **end**
**end**
```
// Check convergence
```
**if** $use\_cosine = True$ **then**
  $centroid\_change \leftarrow 1 - $ mean$(\sum_{j=1}^{embed\_dim} centroids[i,j] \cdot new\_centroids[i,j])$;
**else**
  $centroid\_change \leftarrow \|new\_centroids - centroids\|$;
**end**
$centroids \leftarrow new\_centroids$;
**if** $centroid\_change < tolerance$ **then**
  **break** ;                                              // Converged
**end**
Compute final statistics and coherence scores;
**return** $cluster\_ids, centroids$;

Table 8: Example words of clusters. Shown is the #cluster=32 example.

| Cluster | Size | Example Words |
|---|---|---|
| 1 | 862 | aign, York, Washington, San, Calif, California, Tex, London, Texas, college, ville, football, England, icago, eah, Florida, Chicago, NFL, Jose, ijuana, Los, Louis, Angeles, Carolina, Virginia, Francisco, Boston, Lake, Toronto, Penn |
| 2 | 2098 | ility, ution, ism, ness, formation, ience, effect, result, iness, ency, interest, ality, design, view, cost, story, claim, ization, example, urity, idence, reason, ulation, ability, question, love, problem, success, erence, value, experience, mind, future |
| 3 | 217 | 2016, 2015, 2014, 2017, 2013, 2012, 2011, 2018, 2010, 2008, 2009, century, 2007, 2015, decades, 2000, 2006, 2014, 2005, 2016, 2004, 2017, decade, 2003, 2001, 2002, 1990, 2013, 2012, 1980, 1999, 1998, 2018, 1970, 1996, 1997, 1995 |
| 4 | 776 | country, Euro, Europe, North, South, countries, German, Israel, China, Japan, Syri, Russia, Canada, Russian, European, UK, British, English, Muslim, India, Iraq, Indian, Chinese, Iran, istan, French, Syria, Germany, Arab, Australia |
| 5 | 1746 | ments, system, art, things, information, ways, list, data, plan, process, program, project, money, product, ories, itions, event, games, access, amount, series, ources, hours, space, period, ients, item, ilities, issues, minutes, changes, results |
| 6 | 1886 | God, character, film, King, Dragon, Star, Super, craft, battle, uild, Earth, Game, Dark, movie, ternal, Lord, enemy, Rock, spell, Edition, Cat, Sky, novel, magic, Dead, Master, god, enemies, Games, Hell, Death, Battle, quest, Magic, Wars |
| 7 | 1381 | erson, ason, Trump, oney, son, ohn, augh, ague, erman, inton, Obama, Clinton, White, Hill, ston, wood, ford, berg, Rich, Smith, Brown, sey, zen, bert, Johnson, kin, kins, Fox, Jones, Bush, igan, rick, stein, Lee, Sanders, sen, mond, Williams |
| 8 | 948 | Ed, John, Russ, Mark, David, Paul, Sam, Jack, James, Donald, Rob, Michael, Angel, Ben, Bill, William, Tom, Thom, Alex, George, Phil, Matt, Scott, Lou, Mike, Tim, Hillary, Carol, Chris, Robert, Frank, Mary, Jim, Max, Jeff, Ann |
| 9 | 1790 | ited, made, used, found, read, met, told, ished, got, called, pped, able, ived, available, took, done, seen, thought, given, won, known, sent, based, asked, started, taken, added, reported, itted, wanted, saw, began, overed, needed, lost |
| 10 | 1033 | also, only, again, even, now, then, very, But, ually, still, though, really, fact, ically, too, never, ately, ably, So, always, ently, already, least, later, enough, sure, actually, ever, often, course, yet, likely, today, ially, particular, anything, though |
| 11 | 1064 | war, attack, sex, death, milit, cult, demon, terror, arrest, attacks, Islam, society, violence, prison, ban, crime, peace, igration, criminal, intelligence, assault, politics, marriage, crisis, abuse, murder, killing, bomb, gender, icide, illegal |
| 12 | 2175 | Contact, Content, Truth, Memory, Thread, Eye, Info, Quality, Too, Active, Single, Following, Friends, Ord, Rules, Drop, yrics, Until, Between, Pred, Rule, Weight, Comb, Learn, Inside, Basic, Site, Sex, Normal, Upon, Due, Made |
| 13 | 1764 | ear, car, leg, bus, camp, cap, book, air, dam, land, area, net, room, object, house, field, pub, lab, bar, ground, pack, foot, ograph, page, gun, port, track, heart, ball, building, inside, card, base, hands, eyes, road, goal, phone, table, box, path, pen |
| 14 | 1422 | 4, 5, 6, 7, 8, 9, 00, 01, 201, 19, 6, 7, 8, 000, 200, 10, 12, 12, 50, 19, 11, 20, 18, 201, 30, 15, 15, 11, 14, 16, 16, 14, 13, 25, 18, 13, 30, 17, 199, 17, 25, 24, 80 |
| 15 | 2314 | H, L, E, J, K, St, Th, Y, Ch, irst, Un, Re, Tr, Wh, Sh, Ar, Com, New, Al, Se, Le, Cl, Americ, De, An, Bl, Z, Ad, Fr, Sp, Pl, Be, Ph, Ind, Sc, Or, Con, Comm, Mar, Dr, Fl, Col, Sy, Te, Am, Br, Pr, Pres, Co, Mr, Su, Tw, Ob, Im, Min, Man, Res |
| 16 | 1628 | add, see, because, make, where, against, help, follow, find, govern, ize, during, develop, without, allow, using, ask, invest, build, including, keep, kill, why, tell, give, expect, perform, within, making, become, having, elect, sever, whether |

Table 9: Example words of clusters (Cont.). Shown is the #cluster=32 example.

| Cluster | Size | Example Words |
| --- | --- | --- |
| 17 | 1535 | go, get, play, act, bet, look, don, think, start, count, take, going, run, lead, pass, turn, something, put, might, feel, must, come, seem, stand, let, rest, pay, didn, return, move, pop, talk, until, win, doesn, came, happen, sit, appear, grow |
| 18 | 1043 | K, M, N, Q, U, X, C, U, V, Q, US, X, AT, IS, AS, IT, IC, AC, ST, AM, OS, BC, AD, US, EC, AP, AM, PM, IS, OP, TV, BI, AP, IG, AN, IA, ST, EM, FL, PS, OD, AA, RA, IM, II, IP, SA, CC, CT, PC, SP, BA, BS, OC, ID, NA, PA, GB, IR |
| 19 | 3630 | fter, ople, nder, ause, iew, ublic, ange, stem, vern, ollow, ween, alth, illion, ption, nce, ittle, chool, erest, lease, ording, chn, reen, arge, ww, ample, uro, iven, raph, ission, ember, orth, ajor, outh, ither, arent, ccess, nect, iversity, ideo, gin, ource |
| 20 | 1993 | clud, includ, differe, requ, somet, velop, exper, offic, expl, comple, belie, incre, contin, riend, commun, adv, ruct, beh, ide, polit, ploy, uthor, chang, charact, techn, rese, ocr, econom, orig, polic, creat, ident, invol, rele, ortun, viol, sugg |
| 21 | 1205 | rub, cann, ast, bull, wal, suc, phen, lit, af, vac, upper, cro, sav, lob, root, nav, mand, buff, mo, du, flex, medium, gro, stim, pow, ped, pra, ble, bes, tor, pip, sho, CO, hyd, sup, vent, rum, spr, poly, cock, ju, vo, tort, li, pref, mel, blind, sin |
| 22 | 1714 | app, irect, import, ware, htt, type, function, version, init, file, text, code, method, log, webs, link, htt, struct, tool, search, Twitter, Google, email, command, Facebook, atform, (), application, print, coin, computer, software, feature, Windows |
| 23 | 2353 | Bas, Rad, Incre, Build, Team, Keep, Cap, etition, Real, Disc, Product, Que, King, Tra, Pan, Index, Sk, June, Bro, Looking, English, Vis, Intern, Field, Tor, Microsoft, Rober, Prov, Obama, Supp, Control, Sun, Home, Hub, Apple, Notes |
| 24 | 1217 | ational, ative, ular, public, ural, health, ional, human, itional, social, local, political, individual, anal, ederal, custom, national, aterial, white, military, ental, energy, istic, personal, atural, online, legal, ological, ancial, private, sych, civil |
| 25 | 1236 | ATES, PAR, APTER, DAM, URES, VERY, AMES, GOD, WORLD, HOW, LOC, ITS, REL, VIS, AIN, ATIONAL, CLE, FINE, BEL, IMP, SOU, PUT, ZZ, INC, IVERS, OUR, CLE, LOT, DOWN, TAIN, GER, ITIES, LED, NET, STE |
| 26 | 1427 | report, law, state, government, Rep, ology, conom, school, United, business, season, police, tax, official, city, press, War, State, fund, States, market, campaign, study, community, media, University, Republic, House, service, party |
| 27 | 2042 | very, round, day, self, the, view, ertain, stand, ready, for, not, pro, most, play, empt, urther, wh, year, put, now, minist, load, ext, like, cle, medi, su, time, key, face, rect, set, body, action, ruction, ground, order, down, hold, come, where |
| 28 | 1751 | much, those, such, end, ting, long, ef, ys, its, ax, own, set, ife, ble, ward, show, ms, omet, say, ts, ful, und, ren, cess, ince, tt, olog, up, ump, last, ures, ars, ues, cy, io, hes, air, ier, read, ank, atch, ever, point, ork, ool, alk, ement, ract, same |
| 29 | 1717 | new, good, great, different, fun, big, direct, better, small, vious, current, ailable, large, important, clear, certain, possible, proper, complete, young, strong, quick, true, bad, similar, entire, specific, previous, recent, significant, difficult, easy |
| 30 | 1381 | meric, water, iol, pot, food, wind, pie, drug, blood, oil, gas, brain, fat, gold, birth, drink, skin, cook, disease, chem, plant, species, animals, ice, cells, cohol, cancer, heat, drugs, fuel, diet, alcohol, coal, marijuana, ffee, dry, fish, egg, temperature |
| 31 | 1308 | man, people, ists, person, team, child, men, women, friend, ians, family, company, children, author, intern, others, members, president, players, former, President, mom, someone, player, cop, woman, students, exec, friends, everyone, parent |
| 32 | 1611 | !, #, $, (, ), *, +, ;, <, =, >, ?, @, [, ], _, ', \|, , –, .., ., -, www, http, HAHA |

# D Unconditional Generation

We finally provide some examples of unconditional generation. We choose our HDLM-128-base model, and use the reverse process as described in the main text.

| Sampling Steps | Contents |
| --- | --- |
| 512 | It seems that HTC (TM) is handing a part responsibility to build Smart Device Projects to Qualcomm, concluding the giant former handset partnership, and the 24 Samsung Gear in Japan ordering ESPA+.Check out: Samsung TV with P35 and Sony Glass? Rooster. Samsung aforementioned Desire 1520 was launched last month, and Samsung is now using the LG Desire moniker, the only company to come in its first series of ARM chips powering the 1520 HDD screen. However, a well estimate of the upcoming Gear tells the Beyond Android forums that Samsung users will get access to its Mobile Gear program with each monthly purchase, perhaps it an acknowledgement to its ad-init nature of SME lock, and that Samsung is already shooting a track for further marketing after the device has introduced. only started engineers here, next steps are in Desire 1520 and not in the HTC board. Sale pointed its backers, working on the project, much more up the supply chain, this with the plans for Galaxy launch in 2016 are taken seriously in pre-2015 where Samsung refers a higher end hardware pack of chips to see Wireless Design tests on touchscreen tablets. and their market still beyond the successor, all Galaxy Note 3. Many were surprised that Samsung will not pursue Mobile Fusion during the coming initiative and Strategy period, perhaps because of the flexibility of bringing in such slab of HGDDR, where development is under pressure from certain suppliers. Native research laboratories within the companies would vary some be al possible which manufacturer and OEM has the best push to stay in research lane. |
| 1024 | It opened so slowly, revealing that I am an angry, serious person. I will be typing to end "need you to send me a wallally quick enough? Hopefully it can bring people tears and hurt" psuedo for the real impacts of a heavy hand on my feelings. Unfortunately, I will often be sent a truly open letter long time surprisingly few people ever answer it and take it seriously. Nor besides people tell me I am doing anything wrong. Do you ever sit assured where I would think later? (mathing) Yes, I am talking to you for the first time. I believe it too fast. Mingya Dear blood. Why did you have to read this written hard time? In a desimimized way, you got beaten 3-0 by drinking alcohol against a Zerg strager in our fan club where you now yours. It and the irony is a bit unusual, so many times, I would not check Reddit and accept I was on the other side. In fact, I used to get that kind of Ape feel pulling to let me know. But not now. We are outside of Korea where players do not even know that violence punishment breaking during the game saying Thanks, you can never expose your friends or people to parasitic infections of said hate. I tried somewhere there is that sort of vi-Re where agents fight like "in takes the bullet" and hits back groups but now this Battle is that same kind of place. MTSong: Thanks to my close friends as my translator. I will see you soon. Zona Princess -uncredential -talk -beautifully Braid: When Baran's fan got forthright about your backstabbing and he tried to insult you in intimidation, what made you realize last minute? |

| Sampling Steps | Contents |
| --- | --- |
| 1024 | Almost literally mission based on defying celestial navigation rules making observer looking at objects from 3 bright wavelengths by a century convention. Think of this image as a Gaussian filter, a blur between Galileo and Zeeb, wondering what the objects are going through from the point down. Its stray segments, orbits, orbital motions, unique orbital lengths, all normal events that seem impossible for astronomers to predict. This is just the way the Discovery team wanted the modern astronomy to look, where even the telescope looks up at the sky as it pummels it instead of measuring a star. These objectologist smell like liar on the chaos and like cheat astronomer. So we know the test that followed a unionized framework at Southwest Research Division focused on sending up the groundwork to get a reality instrument to the ground that could in a decade or more better upgrade the world astronomical tools. But they did so largely by accident. The Milky Way provided the Planetary Systemector naked and sent tonnes messages through the solar system, all along way) to the moon. This culture has extended creatures experienced from dusty cloud lifeworm, Cream Dwarf B (or or B or O) to the solar system forgotten white dwarf, Star Power Letter Flake Feeder. Consequently, the helloubating, a deafening cheer, this message was focused on categorizing the minor planets on earth. Somewhere near the gastrarian referred to the message as "Malmy" and running that is like if Lady Orr opens whats away message she has written asking, Everybody belong, go spaceship, I'm working along the planetary science studying various planets on this planet around me, the resulting announcement was useless for most of the carrier. The remaining things just were moved to stop others from moving around before they heard the signals. Everybody's RB, the broadcast is not a observatory to reach for aliens, we're thinking more about it and the message's message seemed supposed to be released to verify a alien's structure via the 1996 SETI conference. Source: Astrophysicinos Gramento de Mancera, et al.- PSA Space Center. G flipped from the corrected message box shows GAnd literally, the probe was supposed to give us information. The entire experimentation mechanism that processed the message was immediately sent back to the NASA Data Center. It would be grand to know the "Daily message box" was off a hoax that this scoring point from a culture. |
| 1024 | One sunny Sunday we ventured to Montreal from over Britannia: when we finally arrived in town the talk about the way of the other day on the rink was the focus. The couch was mixing pie, savanna and chocolate pizza. I was eating bargain thin desserts in my almost-Frances de Don diet on Sunday morning. Now instead of getting homage at the rink, I was with a milkshake hint in the form of caramel cookie baked and deep-fried corn. There were four friends at work at a restaurant. Teddy, my wife and I left sans bread but a table great full house full of work and we leaned in at scrambling. I provide replacements for these-u.do.-e-but-very-krelli-moress, despite Fireade bacon and cheese reheat too far through the crust when we baked the crack. Given the size of my family, my parents stepped in to buy the cookies, break something about baking. Not only that, but we spilt spaghetti-drambled aore through the flame crust and the best bill cooled and sheathed. The candy is sensitive, the bread is thin and it was wrong, it worked. Toated it with my hand. Shut the doors and chew with my fingers. Iated a tail. To the couch while the kitchen pocked and, sure enough, what I thought was a pot was the grilled cheese. Only to be wrong. The cheese had become too dense with grease. |

