# OpenReview forum: "Next Semantic Scale Prediction via Hierarchical Diffusion Language Models"
_NeurIPS.cc/2025/Conference — NeurIPS 2025 poster_

### Official Review · Reviewer_mLoq · 2025-06-29

**Clarity:** 2
**Significance:** 3
**Originality:** 3
**Rating:** 4
**Confidence:** 2

**Summary:**

This paper introduces Hierarchical Diffusion Language Models (HDLM), a discrete diffusion framework for language modeling.It structures tokens into semantic hierarchies where detailed tokens map to coarse-grained ancestors. HDLM's forward process independently transitions tokens to higher abstraction levels via scheduled diffusion. The reverse process progressively predicts finer semantics through time-varying next-scale prediction.This enable tokens to occupy different semantic levels at each diffusion step. Theoretically, HDLM derives a closed-form ELBO showing it generalizes prior discrete diffusion models, introduces stochastic perturbations for self-correction, and enables flexible training schemes. Experiments demonstrate HDLM outperforms existing diffusion models in language modeling perplexity and downstream language understanding tasks while requiring less training data, with larger semantic hierarchies yielding stronger self-correction capabilities.

**Questions:**

- Can other models not compute the self-corrected perplexity for unified comparison?
- Did the authors conduct ablation studies on the tricks for HDLM mentioned in the aforementioned drawbacks? Presenting these would more convincingly validate the effectiveness of these designs.
- It is recommended to elaborate on the specific clustering algorithm and operations in the paper, as I believe this is a crucial step for HDLM, directly impacting the model's final performance.
- Adding noise to word vectors in the embedding space causes the perturbed vectors to follow a certain probability distribution around their original positions. Does this correspond to the transformation from precise semantics to coarse-grained semantics in the proposed method?

**Ethical Concerns:**

["NO or VERY MINOR ethics concerns only"]

**Final Justification:**

raise score to 4

**Limitations:**

yes

**Quality:**

3

**Strengths And Weaknesses:**

Strengths：
Proposes a hierarchical diffusion framework (HDLM) that introduces semantic hierarchies between word-level tokens and mask tokens. This replaces uniform/mask noise with coarse-to-fine semantic transitions, addressing key limitations in prior discrete diffusion models. Theoretical innovations include deriving closed-form ELBO expressions for hierarchical diffusion and proving HDLM generalizes existing models HDLM’s hierarchical design enables error mitigation during decoding, validated by significant gains in self-corrected perplexity. Techniques like stochastic perturbations and force transition improve robustness and training stability. HDLM achieves state-of-the-art self-corrected perplexity among diffusion models and consistent improvements over the baseline method GIDD across all datasets. The schematic diagram is clear and easy to understand.

Weaknesses:
Although Figure 1 is clear and easy to understand, the methodology section of the paper is rather obscure and suffers from poor readability.
The experimental section is insufficient, with only limited comparisons of perplexity and downstream task performance against a few methods. The experimental setup lacks clarity, particularly regarding implementation details such as the clustering process. Additionally, the authors did not investigate the model's performance across more hierarchical levels.

Furthermore, while the paper proposes several tricks for HDLM, including Stochastic Perturbation \& Self-Correction, Hard Training Mode, Auxiliary Loss, and Force Transition, the experimental section fails to provide ablation studies for these components. This omission makes it difficult to assess their necessity and individual contributions to the model's performance.

---

> ### Author Rebuttal · Authors · 2025-07-31
>
> Thank you for your thoughtful review. We have improved our papers accordingly, and hope that we could address your concern.
>
> > Although Figure 1 is clear and easy to understand, the methodology section of the paper is rather obscure and suffers from poor readability.
>
> * We will improve the writing by providing more intuitive interpretations over the extensive theory, and explain the method in detail according to Figure 1.
>
> > The experimental section is insufficient, with only limited comparisons of perplexity and downstream task performance against a few methods.
>
> * We conduct additional experiments by training HDLMs wiith different number of clusters and compare both validation perplexity and generative perplexity metrics with more baselines. For generative perplexity, we generate 256 sequences with 512 length using 512 denoising steps and leverage gpt2-large as the reference model. The new results validate that our HDLM substantially outperform baseline models MDLM and GIDD in terms of both validation perplexity and generative perplexity. In addition, we train HDLM on the LM1B datasets and shows that our method outperforms previous baselines such as MDLM and SEDD.
>
> Table 1: Performance on OWT dataset.
>
> | Model (small)  | train tok | valid ppl ($\downarrow$) | gen ppl ($\downarrow$) |
> | -------------- | --------- | ------------------------ | ---------------------- |
> | GPT2           | unk.      | 23.40                    | -                      |
> | SEDD           | 262B      | $\leq$24.10              | -                      |
> | MDLM (reimpl.) | 131B      | $\leq$27.48              | 170.7                  |
> | GIDD (reimpl.) | 131B      | $\leq$25.71              | 170.2                  |
> | HDLM-1         | 131B      | $\leq$25.72              | 163.9                  |
> | HDLM-2         | 131B      | $\leq$25.79              | 160.2                  |
> | HDLM-4         | 131B      | $\leq$25.05              | 162.0                  |
> | HDLM-8         | 131B      | $\leq$24.57              | 146.6                  |
> | HDLM-16        | 131B      | $\leq$24.17              | 155.2                  |
> | HDLM-32        | 131B      | $\leq$23.83              | 154.6                  |
> | HDLM-64        | 131B      | $\leq$23.36              | $\textbf{144.2}$                  |
> | HDLM-128       | 131B      | $\leq$$\textbf{23.25}$              | 148.0                  |
> | HDLM-256       | 131B      | $\leq$23.65              | 150.4                  |
> | HDLM-128-Large | 131B      | $\leq$ $\underline{19.22}$              | $\underline{139.9}$                  |
>
> Table 2: Performance on LM1B dataset.
>
> | Model                       | PPL ($\downarrow$) |
> | --------------------------- | ------------------ |
> | AR Transformer (33B tokens) | 22.32              |
> | BERT-Mouth                  | $\leq$142.89       |
> | D3PM (absorb)               | $\leq$76.90        |
> | Diffusion-LM                | $\leq$118.62       |
> | DiffusionBert               | $\leq$63.78        |
> | SEDD (33B tokens)           | $\leq$32.79        |
> | MDLM (33B tokens)           | $\leq$27.04        |
> | HDLM-64 (33B tokens)        | $\leq$$\textbf{26.95}$        |
>
> * It is worth mentioning that our experimental settings are standard ones that other papers (such as MDLM and GIDD) also deploy, testing only on these metrics.
>
> > The experimental setup lacks clarity, particularly regarding implementation details such as the clustering process.
>
> > It is recommended to elaborate on the specific clustering algorithm and operations in the paper, as I believe this is a crucial step for HDLM, directly impacting the model's final performance.
>
> * Most of our experiments follow the settings of previous work. We explained the experimental settings and implementation details in the Experiment section as well as in **Appendix D**. Regarding the clustering process, we included extensive details in **Appendix E**. Please let us know if you have more questions. Moreover, in our new experiments, we also conducted ablations on the number of clusters, showing that a cluster number of 32, 64, or 128 leads to optimal performance.
>
> > Additionally, the authors did not investigate the model's performance across more hierarchical levels.
>
> * If here "more hierarchical levels" refer to more cluster numbers, our new experiments provide ablations on the cluster numbers with up to 256 clusters. If it means more than one intermediate hierarchies, we already mention this in our limitation section. While we leave the experimental implementation as future work, we derive the novel ELBO for arbitrary hierarchical levels as our new theoretical contribution, which we will state and prove formally in the revised version. We briefly state the main conclusion here.
>
> *Suppose the clean tokens are the $0$-th hierarchy, and the mask token is the $n$-th hierarchy. Denote $c_i(x)$ as the mapping of a clean token $x$ to the $i$-th hierarchy. Let $\alpha_{t,i}$ be the marginal schedule for the $i$-th hierarchy. Then the ELBO is given by:*
>     $$
>     \log p(x)\geq \mathbb E_{t, z_t}\Big[\sum_{i=0}^n -\delta_{z_t, i}\frac{\sum_{j\leq i}\alpha_{t,j}'}{\alpha_{t,i}}\log\Big(\frac{\sum_{x':c_{i-1}(x')=c_{i-1}(x)}p_\theta(x')}{\sum_{x'':c_{i}(x'')=c_{i}(x)}p_\theta(x'')}\Big)\Big]
>     $$
>     where $\delta_{z_t, i}$ is an indicator function that denotes whether $z_t$ is in the $i$-th hierarchy. The result shows that the ELBO of arbitrary hierarchies can be decomposed into a weighted combination of cross-entropy losses on each hierarchy. We believe this is a **novel and highly nontrivial** theoretical contribution which could inspire more future work. Also as we explained to Reviewer UHEs, one intermediate hierarchy may already be sufficient for the language modeling task we are interested in.
>
> > Furthermore, while the paper proposes several tricks for HDLM, including Stochastic Perturbation \& Self-Correction, Hard Training Mode, Auxiliary Loss, and Force Transition, the experimental section fails to provide ablation studies for these components. This omission makes it difficult to assess their necessity and individual contributions to the model's performance.
>
> > Did the authors conduct ablation studies on the tricks for HDLM mentioned in the aforementioned drawbacks? Presenting these would more convincingly validate the effectiveness of these designs.
>
> * Thanks for the suggestions. We agree with the reviewer that more ablation study is needed. While we have already provided the ablations on the cluster number, other ablation experiments are still being conducted, since it takes about five days to train a small size discrete diffusion model on OWT dataset with 8 A100 GPUs. We appreciate your understanding and will report the full results in the following discussion period as soon as the experiments are ready.
>
> > Can other models not compute the self-corrected perplexity for unified comparison?
>
> * We are sorry that we might not understand your question accurately. As a rough answer, the models marked with "x" (including all AR models and masked diffusion models like SEDD, MDLM) are not able to self correct. In comparison, the self-correction ability of HDLM improves its sampling quality.
>
> > Adding noise to word vectors in the embedding space causes the perturbed vectors to follow a certain probability distribution around their original positions. Does this correspond to the transformation from precise semantics to coarse-grained semantics in the proposed method?
>
> * According to previous papers on diffusion language models, the continuous embedding space is hard to control, interpret and generate compared to discrete counterparts. Adding noise to word vectors may result in representations that are close to other tokens, which does not necessarily result in coarse-grained semantics. From a generative perspective, the Gaussian noise in the embedding space actually simulates the (continuous form of) uniform noise diffusion rather than masked diffusion. In contrast, our discrete formulation of intermediate states has clear meanings and definitions for semantic clusters, which is not the case for continuous embedding. Moreover, our coarse-to-fine hierarchies are introduced in an additional vertical dimension (abstract semantics), while adding noise to the embeddings is still operating on the original vocabulary dimension.

---

> > ### Comment · Reviewer_mLoq · 2025-08-05
> >
> > Thanks for the feedback and I will update my scores.

---

> ### Author Response · Authors · 2025-08-05
> **Thank you**
>
> Dear Reviewer,
>
> We sincerely thank you for your response and we are glad that we have addressed your concerns. We will include these new results and improve the paper according to your constructive feedback in the camera ready version.
>
> In addition, we would like to present our new results regarding **Stochastic Perturbation, Force Transition and Self-Correction**.
>
> * We derive the ELBO for the case with random perturbations to the cluster tokens (i.e., $\xi<1$).
>     \begin{align}
>     \log p(x)\geq \mathbb E_{t, z_t}\Bigg[& \delta_{z_t, m}\frac{\beta_{t,m}'}{\beta_{t,m}}\bigg(\xi\cdot\log\Big(p_\theta(c(x))+\frac{1-\xi}{\xi(N_c-1)}[1-p_\theta(c(x))]\Big)\\\ &+\frac{1-\xi}{N_c-1}\cdot \sum_{z_s=c\setminus c(x)}\log\Big(\frac{\xi(N_c-1)}{1-\xi}p_\theta(z_s)+[1-p_\theta(z_s)]\Big)\bigg)\\\ &+\delta_{z_t, c(x)}\frac{-\alpha_t'}{\beta_{t,c}}\log\frac{p_\theta(x)}{p_\theta(z_t)+\frac{1-\xi}{\xi(N_c-1)}[1-p_\theta(z_t)]}\\\ &+\delta_{z_t, c\setminus c(x)}\frac{-\alpha_t'}{\beta_{t,c}}\log\frac{p_\theta(x)}{\frac{\xi(N_c-1)}{1-\xi}p_\theta(z_t)+[1-p_\theta(z_t)]}\Bigg]
>     \end{align}
>
> * We train a small size HDLM-64 with the stochastic perturbation mechanism ($\xi=0.9$) for 1M steps. While the validation perplexity is similar to HDLM-64 without perturbations ($\xi=1$), we observe significantly lower generative perplexity with force transition scheme (**$60\%$ reduction compared with MDLM**). We attribute this success in real inference to the robustness of the model to inaccurate context - with the stochastic perturbation in the training, it is capable of denoising even when some cluster tokens in the context are inaccurate. We opt not to force transition within the cluster during training (so that the model is able to distinguish inaccurate cluster tokens and self-correct), yet force transition at the inference time significantly reduces generative perplexity, as it can minimize the probability of sampling irrelevant tokens. **This impressive improvement strongly validates the effectiveness of the stochastic perturbation mechanism and inference-time force transition scheme, indicating the strong self-correction ability of HDLM.**
>
> | Model (small)       | valid ppl ($\downarrow$) | gen ppl ($\downarrow$) | gen ppl (w/ force transition) |
> | ------------------- | ------------------------ | ---------------------- | ----------------------------- |
> | MDLM (reimpl.)      | $\leq$27.48              | 170.7                  | -                             |
> | GIDD (reimpl.)      | $\leq$25.71              | 170.2                  | -                             |
> | HDLM-64 ($\xi=1.0$) | $\leq$23.36              | 144.2                  | 144.2                         |
> | HDLM-64 ($\xi=0.9$) | $\leq$23.54              | 183.96                 | **69.76**                     |

---

### Official Review · Reviewer_M7sG · 2025-07-02

**Clarity:** 2
**Significance:** 3
**Originality:** 3
**Rating:** 4
**Confidence:** 3

**Summary:**

This paper introduces an innovative class of discrete diffusion models called Hierarchical Diffusion Language Models (HDLM). In essence, HDLM employs a hierarchical vocabulary structure where lower-level tokens convey more detailed semantics, while higher-level tokens offer a more coarse-grained semantic perspective. The authors present a robust theoretical framework alongside practical experiments to convincingly demonstrate the efficacy of HDLM.

**Questions:**

- When applying HDLM to image generation using discrete tokens generated by VQ-VAE, what modifications are necessary? In this context, would HDLM be a suitable choice?
- Why do you use different numbers of clusters for different model sizes in Table 2? Should the number of clusters be fixed before training, and how is this number determined?
- Why did you choose the DiT architecture over the more commonly used LLaMA or Qwen in LLMs? Could you compare HDLM's performance with the LLaMA architecture?

**Ethical Concerns:**

["NO or VERY MINOR ethics concerns only"]

**Final Justification:**

The authors answered all my questions. I still vote for acceptance.

**Limitations:**

Yes.

**Paper Formatting Concerns:**

No.

**Quality:**

3

**Strengths And Weaknesses:**

**Strengths**
- This paper is well-written, and the concept of HDLM is both novel and intriguing.
- The theoretical results and practical design principles are robustly established.
- With its self-correction step, HDLM outperforms all other discrete diffusion models.

**Weaknesses**
- The related work section lacks organization.
- In Table 1, the comparison of HDLM's sampling speed with different cluster numbers against other discrete diffusion and AR models is unclear. Additionally, does the self-correction step introduce extra computational costs?
- Why does the small HDLM outperform the base HDLM in Table 2?

---

> ### Author Rebuttal · Authors · 2025-07-31
>
> We sincerely appreciate the reviewer's positive feedback. We have answered your questions as below.
>
> > The related work section lacks organization.
>
> Thanks for the reminder. We will update the related work section in the revised version.
>
> > In Table 1, the comparison of HDLM's sampling speed with different cluster numbers against other discrete diffusion and AR models is unclear. Additionally, does the self-correction step introduce extra computational costs?
>
> We would like to clarify that the main computation FLOPs occur in the forward pass of DiT, which is the same for HDLM and other baselines (e.g., MDLM) with same architectures. The only extra computation of HDLM is computing the transition matrix between states, which is negligible compared with DiT forward pass. Similarly, the computation is also almost the same for all models with self-correction step, yet it is still efficient since we only need one forward pass (rather than recursively sampling). In summary, HDLM shares almost the same sampling efficiency compared with MDLM in real-inference settings.
>
> > Why does the small HDLM outperform the base HDLM in Table 2?
>
> Actually this phenomenon is also observed in GIDD paper. We attribute this to the gap between the downstream task and the pretraining corpus. Since the base model fits into the pretraining task more than the small model, they may struggle when there is a distribution shift in downstream tasks (namely overfitting). However, we indeed observe significantly better sample quality of base model in unconditional generation tasks, which can be measured by generative perplexity. In detail, for both models with 128 clusters and trained for 131B tokens, we generate 256 sequences with length 512 using 512 denoising steps. When using gpt2-large as the reference model, the generative perplexity is 139.9 for the base model, which is obviously lower than that for the small model (148.0).
>
> > When applying HDLM to image generation using discrete tokens generated by VQ-VAE, what modifications are necessary? In this context, would HDLM be a suitable choice?
>
> This is a great question. Typically, the total number of image tokens - codebook size (e.g., 2048) times hidden dimension (e.g., 1024), is significantly larger than language vocabulary size (typically around 50,000 for gpt2 and around 250,000 for Qwen-2). Also, the visual semantics of image tokens may highly depends on the contexts, whereas language tokens may have relatively static meanings on their owns. These factors make it harder to define proper hierarchies over image tokens. It is definitely reasonable to apply HDLM, and dynamic clustering or modifications similar to existing methods like VAR (which employs next-scale prediction) could make it more suitable. Since the image generation task is beyond the scope of the paper, we leave it as the future work.
>
> > Why do you use different numbers of clusters for different model sizes in Table 2? Should the number of clusters be fixed before training, and how is this number determined?
>
> The number of clusters is fixed as a hyper-parameter. We have conducted extensive ablations on the number of clusters in our new experiments, see the table above. Regarding validation perplexity, we observe substantial improvement when increasing the number of clusters, and optimal performance is achieved when the number of clusters is around 32 to 128. The result is consistent with our intuition: a larger number of clusters would lead to more expressive intermediate states and more sophisticated clusters, making the "mask to cluster'' step harder, yet "cluster to token'' step much easier. Striking a balance between these two steps would improve overall sampling performance.
>
> | Model (small)  | train tok | valid ppl ($\downarrow$) | gen ppl ($\downarrow$) |
> | -------------- | --------- | ------------------------ | ---------------------- |
> | GPT2           | unk.      | 23.40                    | -                      |
> | SEDD           | 262B      | $\leq$24.10              | -                      |
> | MDLM (reimpl.) | 131B      | $\leq$27.48              | 170.7                  |
> | GIDD (reimpl.) | 131B      | $\leq$25.71              | 170.2                  |
> | HDLM-1         | 131B      | $\leq$25.72              | 163.9                  |
> | HDLM-2         | 131B      | $\leq$25.79              | 160.2                  |
> | HDLM-4         | 131B      | $\leq$25.05              | 162.0                  |
> | HDLM-8         | 131B      | $\leq$24.57              | 146.6                  |
> | HDLM-16        | 131B      | $\leq$24.17              | 155.2                  |
> | HDLM-32        | 131B      | $\leq$23.83              | 154.6                  |
> | HDLM-64        | 131B      | $\leq$23.36              | 144.2                  |
> | HDLM-128       | 131B      | $\leq$23.25              | 148.0                  |
> | HDLM-256       | 131B      | $\leq$23.65              | 150.4                  |
> | HDLM-128-Large | 131B      | $\leq$19.22              | 139.9                  |
>
> > Why did you choose the DiT architecture over the more commonly used LLaMA or Qwen in LLMs? Could you compare HDLM's performance with the LLaMA architecture?
>
> One of the fundamental difference between diffusion language model and autoregressive model is that the former features bidirectional attention and is capable of any order generation. These characteristics cannot be achieved by LLaMA or Qwen since they have causal attention and can only do left-to-right sequential generation. Actually, DiT is widely used in continuous and discrete diffusion, as in SEDD, MDLM, GIDD, etc, and we follow their settings. The performance of LLaMA-like model is included in the tables, and it is widely accepted that existing diffusion language models still underperform state-of-the-art AR LLMs, yet the gap is being closed.

---

> > ### Comment · Reviewer_M7sG · 2025-08-04
> >
> > Thanks for your detailed reply. I have no other question.

---

> > > ### Author Response · Authors · 2025-08-04
> > >
> > > Thanks for your positive feedback. We are glad that we have addressed your concerns.

---

### Official Review · Reviewer_UHEs · 2025-07-03

**Clarity:** 2
**Significance:** 2
**Originality:** 3
**Rating:** 4
**Confidence:** 4

**Summary:**

This paper introduces Hierarchical Diffusion Language Models (HDLM), a novel class of discrete diffusion models for language generation. The core idea is to replace the standard denoising process, which typically transitions from a mask token directly to a data token, with a multi-stage, hierarchical process. In the proposed forward process, word tokens are first perturbed into "coarse-grained" semantic clusters before finally becoming a mask token. The reverse process learns to progressively refine these abstract cluster tokens back into specific, "fine-grained" word tokens, a scheme the authors term "next semantic scale prediction". The authors provide a theoretical foundation for this approach, derive an ELBO for training, and show that a previous model, MDLM, is a special case of their framework. They evaluate HDLM on language modeling and downstream tasks, proposing a new *self-corrected perplexity* metric to demonstrate the model's capabilities.

**Questions:**

8.  Could you add the self-corrected perplexity result for an HDLM with 1 cluster (= MDLM) to Table 1?

9.  Given that the ELBO provides only a bound on the true likelihood -- whose tightness can vary between models -- have you considered evaluating the models using a more direct measure of generative quality, such as generative perplexity, which is a standard evaluation in the context of diffusion language models?

10. The paper only considers one intermediate hierarchical level. You mention extending the theory to multiple hierarchies as a limitation. Could you briefly elaborate on the potential benefits and challenges of such an extension?

**Ethical Concerns:**

["NO or VERY MINOR ethics concerns only"]

**Final Justification:**

The authors added new experiments and performance metrics showing improved results. I have thus increased my score and now lean towards acceptance of this work.

**Limitations:**

Yes.

**Paper Formatting Concerns:**

No.

**Quality:**

2

**Strengths And Weaknesses:**

**Strengths**

1. The paper tackles the timely and important problem of improving discrete diffusion models for language. The core idea of introducing semantic hierarchies is novel, intuitive, and offers a principled way to improve the denoising process compared to standard techniques. I find the *next semantic scale prediction* framing a good conceptual contribution.

2. The work is theoretically-grounded

3. The paper demonstrates that its framework is a generalization of previous work, with MDLM being a special case where only one cluster token exists.

**Weaknesses**

4. The paper's primary claim of effectiveness rests on a new metric, i.e., *self-corrected perplexity*. While the motivation to capture the model's refinement ability is understandable, introducing a new metric and using it alone to show your own model's superiority is not fully convincing without external validation. This is especially problematic when the model performs worse on standard metrics and achieves only marginal gains on downstream tasks. Furthermore, perplexity values derived from the diffusion ELBO are an upper bound on the true perplexity, meaning they should technically be reported with a $\leq$ sign. Reporting these as exact values in Table 1 is imprecise.

5. The writing, particularly in the results section (5.2), is not always clear. The justification for the self-corrected perplexity metric is dense and its fairness is just asserted. Key details are also missing; for instance, the notation HDLM-X is used in tables, but 'X' (representing the number of clusters) is not explicitly defined in the captions. The abstract is also vague, using generic sentences like demonstrate the effectiveness without concrete, quantitative details.

6. The paper fails to provide any justification for choosing the number of clusters for different experiments. This is a critical hyperparameter of the proposed method, and its impact on performance, complexity, and other trade-offs is not discussed.

7. The paper frequently uses terms like "detailed semantics", "coarse-grained meanings", and "richer semantics". While the operational meaning (grouping tokens via clustering) is clear, the connection to linguistic semantics is loose and based on intuition rather than a more formal, well-defined concept.

Overall, I think the idea is interesting, yet tested on a single setting and -- currently -- it brings only marginal improvements over simpler baselines.

*Minor*

There are several typos (e.g., L192: "as state" -> "as stated", L262: “yet which does not” -> “yet this does not”?, Table 4: xi -> \xi).

---

> ### Author Rebuttal · Authors · 2025-07-31
>
> We thank the reviewer for the insightful review. We have worked hard to improve our work and answer your questions as below. Hope we have addressed your concerns, and we are happy to discuss any follow-up questions.
>
> > Weakness 4: self-corrected perplexity and other metrics.
>
> * We agree with the reviewer that the original evaluation is somehow limited. While the rationality of self-corrected ppl has already been clarified in the paper, we would like to add that we are not using self-corrected ppl alone as our main metric. We first apologize for accidentally reporting the performance of all our HDLM models trained for 65B tokens (instead of 131B) in Table 1 of the original submission. This insufficient training leads to an unfair comparison with other baselines. Along with moderate hyper-parameter search, including leveraging an improved schedule ($\gamma=1$ in the appendix) and loss weight clipping, we conduct new experiments and train all the models for 131B tokens, yielding **substantially better results on standard metrics including validation perplexity and generative perplexity** compared with MDLM and GIDD baselines retrained in the same setting. Specifically, our best HDLM obtains even lower validation perplexity than gpt-2. The corrected performance of validation perplexity and generative perplexity is updated in the table below.
>
> Table 1: Performance on OWT dataset.
>
> | Model (small)  | train tok | valid ppl ($\downarrow$) | gen ppl ($\downarrow$) |
> | -------------- | --------- | ------------------------ | ---------------------- |
> | GPT2           | unk.      | 23.40                    | -                      |
> | SEDD           | 262B      | $\leq$24.10              | -                      |
> | MDLM (reimpl.) | 131B      | $\leq$27.48              | 170.7                  |
> | GIDD (reimpl.) | 131B      | $\leq$25.71              | 170.2                  |
> | HDLM-1         | 131B      | $\leq$25.72              | 163.9                  |
> | HDLM-2         | 131B      | $\leq$25.79              | 160.2                  |
> | HDLM-4         | 131B      | $\leq$25.05              | 162.0                  |
> | HDLM-8         | 131B      | $\leq$24.57              | 146.6                  |
> | HDLM-16        | 131B      | $\leq$24.17              | 155.2                  |
> | HDLM-32        | 131B      | $\leq$23.83              | 154.6                  |
> | HDLM-64        | 131B      | $\leq$23.36              | $\textbf{144.2}$                  |
> | HDLM-128       | 131B      | $\leq$$\textbf{23.25}$              | 148.0                  |
> | HDLM-256       | 131B      | $\leq$23.65              | 150.4                  |
> | HDLM-128-Large | 131B      | $\leq$ $\underline{19.22}$              | $\underline{139.9}$                  |
>
> Table 2: Performance on LM1B dataset.
>
> | Model                       | PPL ($\downarrow$) |
> | --------------------------- | ------------------ |
> | AR Transformer (33B tokens) | 22.32              |
> | BERT-Mouth                  | $\leq$142.89       |
> | D3PM (absorb)               | $\leq$76.90        |
> | Diffusion-LM                | $\leq$118.62       |
> | DiffusionBert               | $\leq$63.78        |
> | SEDD (33B tokens)           | $\leq$32.79        |
> | MDLM (33B tokens)           | $\leq$27.04        |
> | HDLM-64 (33B tokens)        | $\leq$$\textbf{26.95}$        |
>
>
>  * We appreciate the reviewer's reminder of the $\leq$ sign and will include it in the revised version. We fully understand that diffusion ELBOs are upper bounds, and we have emphasized that the bound for HDLM tends to be looser (as explained in the paper). We omitted the $\leq$ as it should be a common sense and will update Table 1 with the $\leq$ in the revised version.
> \end{itemize}
>
> > Weakness 5: writing
>
> * We will improve the writing to make it clearer. We will make the justification of self-corrected perplexity more concise, explicitly define HDLM-X where $X$ is the number of clusters, and use concrete, quantitative details in the abstract. We will also fix the typos.
>
> > Weakness 6: number of clusters
>
> * We have conducted extensive ablations on the number of clusters in our new experiments, see the table above. Regarding validation perplexity, we observe substantial improvement when increasing the number of clusters, and optimal performance is achieved when the number of clusters is around 32 to 128. The result is consistent with our intuition: a larger number of clusters would lead to more expressive intermediate states and more sophisticated clusters, making the "mask to cluster'' step harder, yet "cluster to token'' step much easier. Striking a balance between these two steps would improve overall sampling performance. Regarding complexity, larger number of clusters would indeed lead to a larger transition matrix definition (namely $\Gamma\in\mathbb R^{|C|\times|V|}$). However, the actual computation overhead is negligible compared to the DiT forward pass.
>
> > Weakness 7: connection to linguistics
>
> * While the strict connection to linguistics is beyond the scope of the paper, it is indeed an interesting question. We present some examples of clustered tokens in the supplementary (Table 5 in Appendix E.2), where one could easily observe patterns (e.g., same part of speech, similar meanings, and shared root or etymology). However, we would like to emphasize that machine learning models may learn in a different way from humans, thus they may benefit from abstract representations (e.g., semantic clusters in this paper). Even the strict linguistic analysis of LLMs is difficult, and more research on the connection between LLMs/dLMs and linguistics is worth exploring in the future.
>
> > Question 8: Could you add the self-corrected perplexity result for an HDLM with 1 cluster (= MDLM) to Table 1?
>
> * Self-corrected perplexity for HDLM with 1 cluster is 25.68, which is almost the same as the original validation perplexity, verifying that MDLM is not able to self-correct.
>
> > Question 9: Given that the ELBO provides only a bound on the true likelihood -- whose tightness can vary between models -- have you considered evaluating the models using a more direct measure of generative quality, such as generative perplexity, which is a standard evaluation in the context of diffusion language models?
>
> * Thanks for the reminder. It is true that ELBO is a bound of the likelihood, and as we emphasized in the paper, the bound for HDLMs may be looser due to the extended vocabulary size, yet they still outperform baselines.
> * Regarding other measures of generative quality, we report the generative perplexity in Table 1 above. For all models, we generate 256 sequences, each with a length of 512 tokens. The number of denoising steps is set to 512, and the reference model is gpt2-large. Our HDLM significant outperform MDLM and GIDD baselines (the best HDLM-64 reduces gen. ppl. by over $15\%$).
>
> > Question 10: The paper only considers one intermediate hierarchical level. You mention extending the theory to multiple hierarchies as a limitation. Could you briefly elaborate on the potential benefits and challenges of such an extension?
>
> * This is a great question. Introducing more hierarchies naturally aligns with the``progressively denoising'' natural of diffusion models, which is also one of our main motivation. Multiple hierarchies allow for more sophisticated control and potentially stronger self-correction abilities, since intermediate hierarchies with different semantic levels are capable of providing useful information that mask tokens cannot provide, while maintaining the ability to deal with uncertainty and potential needs to self-correct. For practical challenges, defining such hierarchies may not be trivial, and classical methods such as hierarchical clustering cannot necessarily lead to optimal definitions of the transitions. As mentioned in the paper, dynamically learning the hierarchies may be a promising direction, which we leave as future work. Furthermore, since common vocabulary sizes of the models are typically 50,000 (such as gpt-2 tokenizer) to 250,000 (such as Qwen-2 tokenizer), one intermediate hierarchy may already be sufficient. More hierarchies would lead to smaller-size clusters and restricted transition matrix, limiting the expressiveness while introducing higher computation costs.
> * While we leave the experimental implementation of more hierarchies as future work, we actually derive the elbo for arbitrary number of hierarchies, which we will state and prove formally in the revised version. We briefly state the main conclusion here.
> *Suppose the clean tokens are the $0$-th hierarchy, and the mask token is the $n$-th hierarchy. Denote $c_i(x)$ as the mapping of a clean token $x$ to the $i$-th hierarchy. Let $\alpha_{t,i}$ be the marginal schedule for the $i$-th hierarchy. Then the ELBO is given by:*
>     $$
>     \log p(x)\geq \mathbb E_{t, z_t}\Big[\sum_{i=0}^n -\delta_{z_t, i}\frac{\sum_{j\leq i}\alpha_{t,j}'}{\alpha_{t,i}}\log\Big(\frac{\sum_{x':c_{i-1}(x')=c_{i-1}(x)}p_\theta(x')}{\sum_{x'':c_{i}(x'')=c_{i}(x)}p_\theta(x'')}\Big)\Big]
>     $$
>     where $\delta_{z_t, i}$ is an indicator function that denotes whether $z_t$ is in the $i$-th hierarchy. The result shows that the ELBO of arbitrary hierarchies can be decomposed into a weighted combination of cross-entropy losses on each hierarchy. We believe this is a **novel and highly nontrivial** theoretical contribution which could inspire more future work.

---

> > ### Comment · Reviewer_UHEs · 2025-08-04
> >
> > I thank the authors for their rebuttal and the additional experiments. Their responses have addressed my initial concerns. As I noted previously, the core idea is interesting, and given the new results, I now lean towards acceptance and will increase my score. I strongly encourage the authors to incorporate all the updates, new discussions, and results from their rebuttal into their manuscript.

---

> > > ### Author Response · Authors · 2025-08-06
> > >
> > > Dear Reviewer,
> > >
> > > We sincerely thank you for your constructive review and positive feedback. We will include these new results and improve the paper according to your constructive feedback in the camera ready version. We are writing to kindly provide some additional results that you could take into account in your final justification if you like.
> > >
> > > Below are our new results regarding **Stochastic Perturbation, Force Transition and Self-Correction**, which demonstrates strong experimental improvements.
> > >
> > > * Theoretically, we derive the ELBO for the case with random perturbations to the cluster tokens (i.e., $\xi<1$). This reduces to the ELBO in the paper when $\xi=1$.
> > >     \begin{align}
> > >     \log p(x)\geq \mathbb E_{t, z_t}\Bigg[& \delta_{z_t, m}\frac{\beta_{t,m}'}{\beta_{t,m}}\bigg(\xi\cdot\log\Big(p_\theta(c(x))+\frac{1-\xi}{\xi(N_c-1)}[1-p_\theta(c(x))]\Big)\\\ &+\frac{1-\xi}{N_c-1}\cdot \sum_{z_s=c\setminus c(x)}\log\Big(\frac{\xi(N_c-1)}{1-\xi}p_\theta(z_s)+[1-p_\theta(z_s)]\Big)\bigg)\\\ &+\delta_{z_t, c(x)}\frac{-\alpha_t'}{\beta_{t,c}}\log\frac{p_\theta(x)}{p_\theta(z_t)+\frac{1-\xi}{\xi(N_c-1)}[1-p_\theta(z_t)]}\\\ &+\delta_{z_t, c\setminus c(x)}\frac{-\alpha_t'}{\beta_{t,c}}\log\frac{p_\theta(x)}{\frac{\xi(N_c-1)}{1-\xi}p_\theta(z_t)+[1-p_\theta(z_t)]}\Bigg]
> > >     \end{align}
> > >
> > > * Experimentally, we train a small size HDLM-64 with the stochastic perturbation mechanism ($\xi=0.9$) for 1M steps. While the validation perplexity is similar to HDLM-64 without perturbations ($\xi=1$), we observe significantly lower generative perplexity with force transition scheme (**$60\%$ reduction compared with MDLM**). We attribute this success in real inference to the robustness of the model to inaccurate context - with the stochastic perturbation in the training, it is capable of denoising even when some cluster tokens in the context are inaccurate. We opt not to force transition within the cluster during training (so that the model is able to distinguish inaccurate cluster tokens and self-correct), yet force transition at the inference time significantly reduces generative perplexity, as it can minimize the probability of sampling irrelevant tokens. **This impressive improvement strongly validates the effectiveness of the stochastic perturbation mechanism and inference-time force transition scheme, indicating the strong self-correction ability of HDLM.**
> > >
> > > | Model (small)       | valid ppl ($\downarrow$) | gen ppl ($\downarrow$) | gen ppl (w/ force transition) |
> > > | ------------------- | ------------------------ | ---------------------- | ----------------------------- |
> > > | MDLM (reimpl.)      | $\leq$27.48              | 170.7                  | -                             |
> > > | GIDD (reimpl.)      | $\leq$25.71              | 170.2                  | -                             |
> > > | HDLM-64 ($\xi=1.0$) | $\leq$23.36              | 144.2                  | 144.2                         |
> > > | HDLM-64 ($\xi=0.9$) | $\leq$23.54              | 183.96                 | **69.76**                     |

---

> ### Author Response · Authors · 2025-08-04
> **Thank you**
>
> Dear reviewer,
>
> Thank you for your reply and increasing the score. Your constructive feedbacks really help in improving our paper and we are glad that we have addressed your concerns. We will definitely include these new results in the camera ready version.

---

### Official Review · Reviewer_7z47 · 2025-07-04

**Clarity:** 3
**Significance:** 3
**Originality:** 4
**Rating:** 4
**Confidence:** 4

**Summary:**

This paper introduces Hierarchical Diffusion Language Models (HDLM), a discrete diffusion model that operate on hierarchical semantic scales. Unlike standard discrete diffusion models that transition between clean tokens and masks, HDLM introduces intermediate semantic hierarchies where tokens progressively transition from fine-grained to coarse-grained representations. The authors derive closed-form ELBO expressions, demonstrate that existing models like MDLM are special cases of their framework, and show that HDLM achieves competitive performance on language modeling tasks while enabling self-correction through its hierarchical structure.

**Questions:**

No

**Ethical Concerns:**

["NO or VERY MINOR ethics concerns only"]

**Final Justification:**

The authors covered some of my concerns in their rebuttal. I believe that this paper is good, yet I cannot give it a rating of 5, mostly because of other unaddressed points in the weaknesses. I believe this paper would benefit from a more in-depth exploration of the underlying representations. I don't think that the fact that other people haven't done that is a good argument.

**Limitations:**

Yes

**Quality:**

2

**Strengths And Weaknesses:**

Strengths:

- Conceptually novel approach: The idea of hierarchical semantic diffusion with "next semantic scale prediction" is intuitive and well-motivated, addressing key limitations of existing discrete diffusion models (lack of semantic richness in masked tokens, inability to self-correct).
- Strong theoretical foundation: The paper provides rigorous mathematical treatment with closed-form ELBO derivations and proves elegant properties like the invariance of token-level loss weights.
- Unifying framework: Showing that MDLM is a special case of HDLM demonstrates the generality of the approach and provides theoretical validation.
- Self-correction capability: The hierarchical structure naturally enables the model to refine predictions, as demonstrated by improved self-corrected perplexity.

Weaknesses:

- Insufficient insights into model behavior: The paper presents only quantitative metrics (perplexity, downstream task accuracy) without providing any understanding of how HDLM actually functions as a language model. Metrics alone are insufficient to understand a generative model's behavior.
- No generation examples or analysis: The paper lacks any examples of generated text, making it impossible to assess generation quality, coherence, or how the hierarchical structure influences output. Readers cannot judge whether the model produces sensible text or how it differs qualitatively from baselines.
- Missing generation process visualization: How does the hierarchical denoising process unfold? Examples showing the progression from mask→cluster→word tokens would illuminate the model's unique generation dynamics and help readers understand the practical implications of the hierarchical structure.
- Lack of parameter sensitivity studies: How do key hyperparameters (number of clusters, clustering methods, noise schedules, force transition) affect generation behavior? Without these ablations, it's unclear which design choices matter and how to configure the model effectively.
- No analysis of learned representations: What semantic patterns emerge in the hierarchical clusters? How does the model utilize the intermediate representations during generation? Such analysis would provide crucial insights into whether the theoretical benefits translate to meaningful semantic organization.

---

> ### Author Rebuttal · Authors · 2025-07-31
>
> We address the reviewer's concerns as follows.
>
> > Insufficient insights into model behavior
>
> In our original submissions, we already included sampled texts as qualitative results in the appendix. In our new results, we also report generative perplexity of generated samples, showing that samples generated by our HDLM are substantially better than other baselines. Regarding theoretical insights, we believe our grounded elbo derivation and other discussions in the paper already provided enough insights of the next semantic scale scheme. We can answer in details if you could specify what type of insights you are expecting.
>
> > No generation examples or analysis
>
> We already included a lot of generation examples in the supplementary. Here are more samples.
>
> - Scientists have noticed that patients include some vascular dysfunction, and report a rapid onset of symptoms and years after giving the procedures to treat this kind of cancer anyway. Cancer more often happens to take effect for tumors, however, lowering median rates for brain tumors as much as the patient requests for curable cancer, resulting in some improvement. Patients started to have blurred episodes as an old friend cancer, and the same seems to be improving everywhere. Patient requests do tend to be history with treatment and have included patients with brain that had mostly been frozen, attached as bone, to have a bone that protrudes in the skin and lumbar defects that wouldn't tear, but they can fracture and cut with no difficulty. Jullk-hootinen has had success in the skin cancers in the breasts and substates such as kidney aisles. During the 1990s, the greatest success was in advanced patients and sub-patients in the development of new angi cells. We were to dedicate those of cancerous cancer to cancerous cells, and render our angi cells the same color and colors as our skin and other blood line tissue such as heart, brains and lungs. Except for the Ktellften, basically what I was using as a target in the 1990s. Studios Research, a program by the Strategic Fund, has been toward the study of degenerative cells called angioglial cells that have a coating of freeids neurons. It takes a lot of investments to make a product that's versatile, reliable and working at a reasonable cost. The company has made a monitor, or magnum, to analyze angi cells through a test tube. Just all your angi cells would live in the right tissues. Why allow them to live if we treat an incurable disease, like a new chemo called telov-thole, or not? I am actively investigating angi cells to help treat this defect, so if you are a multi-pathologist who does a set of alkenomalies, some people would have tumors if they could be treated with angi cell therapy. Also, there is a now-saving catheence drug. What would you do with a cancer so advanced? Doctors are brainstorming that now, is a recent device made by a biopharmaceutical company to help with a cancer, like the telov-thole implant, because it's so sensitive and strong."
>
> - As new and healthier China, the country enjoys trends in sustainable technologies and lifestyles. After seemingly a century when 80 percent would mean an annual case of new and compared postings, the term green it had become the metric system of all sustainability - to define and elevate what and where humans will be. A wave of what we knew could be "We would not expect Xi to rank India's ray of green light as easily as China should achieve under his watch is performing uber statistic". Greenpeace Arevers' Matthew Segius. With collaboration from the America Media and Development Bureau. The earth was silently driven through the river of Mount Beijing - one of four milestones reached.However, the reduction of green as Fuzhou seemed outside of his control. In 2009, China was suddenly on a fire. This afire situation can be reflective of the rapid rate of growth. China's annual GDP ratio more than doubled from less than 7 percent of the population in 2007 or 2008 to more than 14 percent now. In Africa, that same growth of the poor guy benefited with India accounting for the top 3 percent of population as much as the U.S., Africa etc. with population spread percent of that population's income. At a revolution below GDP per person today, everyone in India makes \$4,000 while in China the poorest 50 per cent make that amount. But where human activity seem assured is the tiny concentrations of the photographed, and Xi's comments, while outwardly, were meant to express their probable hopes and aspirations. Green responses It is bizarre and distressing that this paradigm of not "have me or haven't" exists for too long. So much of the global green questions are unfiltered and over 92 percent have already been removed. A return to the close-door meeting where a critical shift for the external, is back unnoticed cautionary message, it comes to numbing heads of all kinds of serious provocations is learning shreds of polluted lakes for right into energy-poor regions, stolen from the Chanolan Monastery, The Great Monsoon Divide, a smuggler in the Singapore nightmare made tapping in Russian meters in Norway for electricity leaks while North Korea seeks for its electricity needs. The fact the questions have been removed from subject relations of grassroots success is key. "While in the future we would like them would be coy if then democratic about that, if successful".
>
> > Missing generation process visualization
>
> Unfortunately, we cannot upload figures to visualize the process. Intuitively, we are capable of both GIDD-type (Bayesian based method) and MDLM-type (analogously to masked diffusion) reverse process, where the mask tokens are gradually transit to clusters and then word tokens according to the time schedule.
>
> > Lack of parameter sensitivity studies
>
> In Table 1 below, we provide extensive ablation study on the number of clusters.
>
> Table 1: Performance on OWT dataset.
>
> | Model (small)  | train tok | valid ppl ($\downarrow$) | gen ppl ($\downarrow$) |
> | -------------- | --------- | ------------------------ | ---------------------- |
> | GPT2           | unk.      | 23.40                    | -                      |
> | SEDD           | 262B      | $\leq$24.10              | -                      |
> | MDLM (reimpl.) | 131B      | $\leq$27.48              | 170.7                  |
> | GIDD (reimpl.) | 131B      | $\leq$25.71              | 170.2                  |
> | HDLM-1         | 131B      | $\leq$25.72              | 163.9                  |
> | HDLM-2         | 131B      | $\leq$25.79              | 160.2                  |
> | HDLM-4         | 131B      | $\leq$25.05              | 162.0                  |
> | HDLM-8         | 131B      | $\leq$24.57              | 146.6                  |
> | HDLM-16        | 131B      | $\leq$24.17              | 155.2                  |
> | HDLM-32        | 131B      | $\leq$23.83              | 154.6                  |
> | HDLM-64        | 131B      | $\leq$23.36              | 144.2                  |
> | HDLM-128       | 131B      | $\leq$23.25              | 148.0                  |
> | HDLM-256       | 131B      | $\leq$23.65              | 150.4                  |
> | HDLM-128-Large | 131B      | $\leq$19.22              | 139.9                  |
>
> Due to the time restriction, we are still conducting experiments on other ablations. We will update the results when they are ready.
>
> > No analysis of learned representations
>
> It is difficult to analyze learned representations and connect them with hierarchical structures. No papers of this type on discrete diffusion or even LLMs have done such analysis. We believe this is beyond the original scope of the paper.

---

> > ### Comment · Reviewer_7z47 · 2025-08-04
> >
> > Thank you. I am updating my score

---

> > > ### Author Response · Authors · 2025-08-04
> > > **Thank you**
> > >
> > > We appreciate the reviewer’s reply and we are happy that your concerns have been addressed. We will certainly include these new results and improve our paper according to your constructive comments.

---

### Note · Authors · 2025-08-12

Dear Reviewers and AC,

We sincerely appreciate your valuable time and constructive reviews. We aim to summarize our paper's contribution and the rebuttal process in this final remark.

We believe this paper is well-motivated, theoretically grounded, and (especially after rebuttal) experimentally effective. During rebuttal, we add extensive new experiments, including improved performance (our models significantly outperform previous baselines), evaluation on more metrics and datasets, and sufficient ablation study. We provide new theoretical results for arbitrary hierarchical levels and stochastic perturbations. We also answer the reviewers' other questions thoroughly, and all reviewers do not convey any remaining concerns. Consequently, all reviewers have already or have promised (including reviewer UHEs who haven't submitted his final justification, which we kindly understand) to increase the score to positive and lean to acceptance.

In our camera ready version, we will take all the reviewers' useful suggestions into account and include all these new results. Thanks again for your efforts in improving the paper.

Sincerely,

Authors

---

### Decision · Program_Chairs · 2025-09-17

**Decision:**

Accept (poster)

**Comment:**

The paper introduces Hierarchical Diffusion Language Models (HDLM), a novel class of discrete diffusion models for language. The core innovation is to replace the standard, one-step denoising process (from a mask token directly to a word token) with a multi-stage, hierarchical approach. The authors provide a strong theoretical foundation for this "next semantic scale prediction" framework, deriving a closed-form ELBO and showing that existing models like MDLM are a special case. This hierarchical structure also uniquely enables a self-correction mechanism during generation.

The strengths of the paper include the novel and intuitive idea of hierarchical semantic diffusion and strong theoretical derivations. Reviewers concerned about the weak experimental results, the lack of ablation studies on key hyperparameters such as the number of semantic clusters and no qualitative analysis. These concerns were addressed by the authors during rebuttal and reviewers reached to a consensus of accepting the paper.

I recommend this paper for acceptance on the condition that the authors incorporate the extensive new experiments, results, and theory described in the rebuttal into the final version.